# NeRV-Diffusion: Diffuse Implicit Neural Representations for Video Synthesis

**Yixuan Ren, Hanyu Wang, Bo He, Hao Chen**[*]**, Abhinav Shrivastava**[*]
University of Maryland, College Park
{yxren,hywang66,bohe}@umd.edu, {chenh, abhinav}@cs.umd.edu
Project page: https://nerv-diffusion.github.io/

## ABSTRACT

We present NeRV-Diffusion, an implicit latent video diffusion model that synthesizes videos via generating neural network weights. The generated weights can be arranged as the parameters of a convolutional network, forming an implicit neural representation (INR) and decoding into videos with frame indices as the input. Our framework consists of two stages: First, a hypernetwork-based tokenizer that encodes raw videos from pixel space to neural parameter space, and the bottleneck latent serves as INR weights to decode; Second, an implicit diffusion transformer that denoises on the latent INR weights. In contrast to traditional video tokenizers that compress videos into frame-wise feature maps, NeRV-Diffusion generates a video as a compact dedicated neural network. This holistic video representation obviates temporal cross-frame attentions while preserving flexible temporal interpolability. The INR decoder and weight latent feature sublinear complexity overhead regarding video resolution and length increase with additional upsampling layers. To enable Gaussian-distributed neural weights with high expressiveness, we reuse the bottleneck latent across all INR layers, as well as reform its weight modulation, upsampling connection and input coordinates. We also introduce SNR-adaptive loss weighting and scheduled sampling for effective training of the implicit diffusion model. NeRV-Diffusion reaches superior video synthesis quality over previous INR-based models and comparable performance to most recent state-of-the-art non-implicit models on real-world video benchmarks including UCF-101 and Kinetics-600. It also achieves outstanding decoding and generation efficiency when scaling up to high-resolution and long videos.

## 1 INTRODUCTION

Video latent diffusion models (LDMs) have achieved impressive generative capability. However, their tokenizers usually inherit from image models and encode videos as individual frame-wise feature maps, ignoring the natural coherence across frames and resulting in redundant representations. Cross-frame attentions (Wang et al., 2023; Guo et al., 2023) are thus introduced to constrain temporal consistency, bloating the model size and leading to massive computation footprint. Moreover, traditional tokenizers have fixed dowmsampling factors, and the latent size will increase quadratically when data resolution doubles. 1D tokenizations Yu et al. (2024b); Wang et al. (2025a) have been explored for holistic latent, while discrete tokens compromise the spatiotemporal granularity.

Implicit neural representations (INRs) are neural networks that fit on single data points. An INR takes unified coordinates as the input and outputs pixels as stored in its model weights. It has shown significant advantages on compression (Sitzmann et al., 2020; Dupont et al., 2021), fast decoding (Chen et al., 2021a), and easy transformation (Mildenhall et al., 2021; Kerbl et al., 2023) by representing data as an integral format of function. The continuity and differentiability of INRs facilitate advanced single-data generative tasks, such as super-resolution, restoration, style transfer and editing, via smooth interpolations within the data space. Its compact representation also contributes to reducing memory overhead, making them highly suitable in resource-constrained environments.

---

[*]Co-corresponding authors.

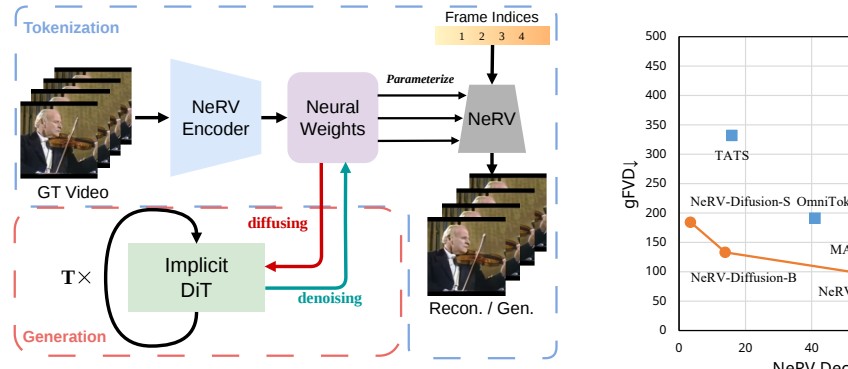

Figure 1: **Left:** Overview of our NeRV-Diffusion framework. In the tokenization stage, NeRV Encoder projects RGB videos to neural weight latent, which serves as the kernels of a NeRV and decodes for reconstruction. In the generation stage, an implicit diffusion transformer is trained to denoise on NeRV weights. During inference, the implicit DiT generates NeRV weights from random noise, and the NeRV decoder renders them into videos. **Right:** NeRV-Diffusion outperforms previous INR-based as well as most recent non-implicit video generation models at all scales with more compact model sizes. The generative performance is evaluated in gFVD on UCF.

To harness the strengths of both latent generative models and implicit neural representations, we establish an implicit latent diffusion model, NeRV-Diffusion, for video synthesis by generating INR weights, where a video is represented as a holistic set of INR weights. It consists of two stages: In the tokenization stage, a hypernetwork-based encoder compresses RGB videos into parametric latent tokens. The tokens instantiate an INR to decode for reconstruction with unified frame indices input. In the generation stage, a diffusion transformer denoises in the encoded implicit latent space, mapping random noise to INR weight tokens. Figure 1 (left) overviews the framework.

However, it is not trivial to acquire Gaussian-distributed neural network weights for smooth diffusion that are meanwhile able to represent diverse realistic data with high fidelity. We adopt a convolutional video INR, NeRV (Chen et al., 2021a), and build a transformer INR encoder based on FastNeRV (Chen et al., 2024a). They are originally designed toward video compression performance only and their produced INR weights are not generatable. To ensure the bottleneck latent tokens fitful for both faithful reconstruction and smooth diffusive generation, we have made several critical architectural modifications. The detailed architectures are illustrated in Figure 2.

Specifically, we reuse the encoded weight tokens with multiple linear affine layers such that each NeRV layer is fulfilled by all tokens independently. We also redesign the weight parameterization approach, proposing to directly set the latent tokens to be the convolution kernels, instead of modulating the shared base weights with scale and shift. These upgrades fundamentally enlarge the expressiveness and smoothness of the implicit space while maintaining its compactness. We leverage vanilla diffusion transformer (Peebles & Xie, 2023) (DiT) to denoise on weight tokens that imply no spatial or temporal structures. We also handle the error accumulation with SNR-adaptive loss weighting and scheduled sampling for optimal denoising in the implicit latent space.

NeRV-Diffusion leverages video INRs as instance-specific decoders, offering faithful reconstruction, compact model and fast decoding compared to the large, shared decoders in traditional LDMs. It encodes and generates video frames holistically as integral INR weights, implies the keyframe-residue representation by reusing the same set of parameters to decode all frames. It thus maintains temporal association and interpolation without cross-frame attention. Furthermore, its weight latent shape is proportional to the INR decoder size, and sublinear to the data resolution upscaling. It only needs to append an additional upsampling block to the decoder to double the output resolution.

In summary, our contributions are as follows:

- We propose a novel implicit video autoencoder that compresses videos into neural weight tokens of normal distribution, constituting generation-specialized video INRs.

- We propose an implicit diffusion model that denoises in neural weight space, achieving dynamic and diverse video synthesis via generating INR parameters.
- NeRV-Diffusion surpasses prior implicit and non-implicit generative models on multiple real-world video benchmarks, and conveys efficient scaling up and smooth interpolation.

## 2 RELATED WORK

### 2.1 IMPLICIT NEURAL REPRESENTATIONS

Implicit neural representations (INRs) are neural networks that fit on single data points. An INR takes in coordinates and outputs corresponding pixel values of the stored data. It has presented capacity and flexibility in various modalities, including images (Sitzmann et al., 2020; Dupont et al., 2021), 3D shapes (Park et al., 2019; Mildenhall et al., 2021) and videos (Chen et al., 2021a; 2022). They are primarily developed for image compression (Strümpler et al., 2022; Dupont et al., 2022) and editing (Fan et al., 2022; Yang et al., 2023), video compression (Li et al., 2022; Kwan et al., 2024; Zhao et al., 2023; Zhang et al., 2021; Lee et al., 2023) and editing (Ouyang et al., 2024), and novel view rendering (Kerbl et al., 2023; Barron et al., 2023; Cao & Johnson, 2023) and 3D scene editing (Yuan et al., 2022; Liu et al., 2024). Although some editing applications have been explored, they create the INRs after manipulating the data in pixel space.

A standard INR is trained via memorizing the pixel data, which is time-consuming in a backpropagation manner. Chen & Wang (2022); Kim et al. (2023a); Chen et al. (2024a) suggest using transformer-based hypernetworks to create INR weights given RGB data in a feed-forward fashion at scale. However, these methods are optimized solely toward reconstruction performance and incorporate no distribution regularization on the produced INR weights, leaving the implicit generative task that synthesizes novel data points from random noise under-addressed.

### 2.2 IMPLICIT NEURAL REPRESENTATION GENERATION

INR generation is a challenging task. Traditional generative models learn mapping random noise to pixels or latent features, while implicit generative models aim to associate neural parameters with Gaussian distribution. Several efforts have been made toward implicit generation. Skorokhodov et al. (2021) builds a GAN for image INRs (Sitzmann et al., 2020), and Yu et al. (2022) extends it to videos by involving the temporal axis.

Erkoç et al. (2023); Chen et al. (2023); Müller et al. (2023); Shue et al. (2023) study generating 3D NeRF parameters via diffusion models. (Chen et al., 2024b) applies latent diffusion models on image INRs (Chen et al., 2021b) for image synthesis, while their INR weights are derived by a complex decoder from the denoising latent space. Recently, Wang et al. (2024b; 2025b) propose to leverage the hypernetwork-INR architecture to conduct flow matching on image or 3D pixel data. Lee et al. (2025) also developed a masked image autoencoder for inpainting with a similar structure. Despite these efforts, no video diffusion model that generates INR weights has yet been explored, casting this a challenging task as videos embed more dynamic information and diffusion models have a more strict demand on its denoising space.

### 2.3 LATENT VIDEO DIFFUSION MODELS

Latent video diffusion models (Wang et al., 2023; Blattmann et al., 2023b; Guo et al., 2023) have achieved significant success in video generative modeling. However, traditional video tokenizers often encode video frames as individual feature maps, calling cross-frame attentions in the denoising network to constrain temporal consistency. Kim et al. (2023b); Wu et al. (2025) start to explore video autoencoders with motion awareness and temporal compression, splitting the complexity between the tokenization and generation stages. Recent 1D tokenization (Yu et al., 2024b; Wang et al., 2025a; Zha et al., 2025) encodes visual data into holistic tokens that project no spatial or temporal alignment with pixels, while they remain focused on images or auto-regressive generation only. In this work, we look to synthesize videos by generating INR weights via diffusion, obviating frame-wise representations by using the whole INR model to decode all frames given time indices. Moreover, symmetric autoencoders rely on a large-scale decoder to render synthesized latent to diverse RGB data with high fidelity, consuming non-negligible computational resources and time for end users

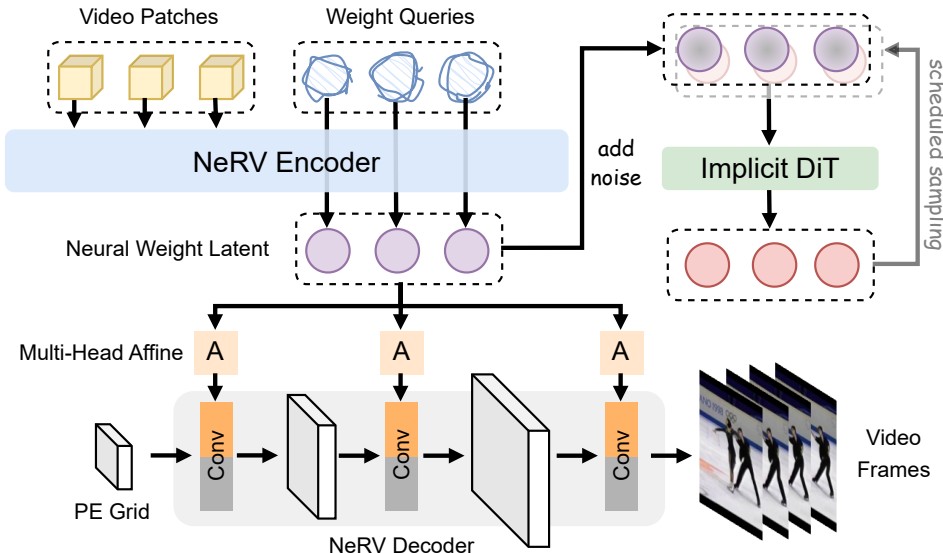

Figure 2: Detailed architectures of NeRV-Diffusion. **Top left**: Patchified videos and learnable weight queries are concatenated and input into NeRV encoder, outputting latent weight tokens; **Middle left:** Weight tokens are reused and converted by multi-head affines to instantiate each NeRV decoder layer; **Bottom:** NeRV decoder renders spatiotemporal positional embeddings into RGB videos, using the instance-specific kernels and global shared kernels. Block details and side connections are omitted; **Top right:** Weight tokens are added noise and an implicit diffusion transformer is trained to denoise in this implicit weight space.

to visualize. We explore the space of asymmetric hypernetwork-INR autoencoders, where the INR acts as an efficient instance-specific decoder as it only needs to represent a single data point.

## 3 NERV DIFFUSION

NeRV-Diffusion is a two-stage generative framework. In the tokenization stage, an implicit autoencoder (§3.1) is trained to compress a video from pixels to latent neural weight tokens, and the tokens function as the parameters of an INR (§3.2) and self-decode to reconstruct the video. In the generation stage, an implicit diffusion transformer (§3.3) is trained to generate the weight tokens from random noise. Figures 1 (left) and 2 illustrate our full pipeline of both stages.

### 3.1 NERV-VAE

In the first stage, we aim to tokenize a video into a latent space that represents the video through the parameters of an INR. This is achieved by training an implicit autoencoder, where the encoder $\mathcal{E}$ is a hypernetwork that produces INR parameters $\theta = \mathcal{E}(x)$ given pixel input $x$. The decoder is implemented as an INR $\mathcal{D}_\theta(\cdot)$, which decodes to pixel values given corresponding coordinates. We build the backbone of our INR encoder $\mathcal{E}$ upon ViT-based FastNeRV (Chen et al., 2024a), where we make several critical modifications to align the learned latent space with generative tasks.

The RGB video is first segmented into patches and converted to transformer input embeddings. Since the output weight tokens have no spatiotemporal correspondence to the input patches, instead of mapping them directly we introduce dedicated query tokens and concatenate them with the data patches following (Peebles et al., 2022). Only the output tokens corresponding to the queries are retained. They are batch normalized along the token embedding dimension.

**KL Bottleneck.** Two additional fully connected (FC) layers are appended after the NeRV encoder's output to create an information bottleneck of compact latent dimension. KL divergence loss is applied to align their distribution toward standard Gaussian distribution $\mathcal{N}(0, 1)$.

**Multi-head Affine Mapping.**   FastNeRV use its encoded latent to modulate the parameters of a subset of the INR layers, which limits the capacity of the latent tokens especially when KL constraint is applied for generation tasks. Inspired by the multiple affine layers in Karras et al. (2019), we expand the post-bottleneck FC layer into multi-head affine mappings, and the single set of weight tokens are reused to modulate all NeRV layers independently. Specifically, for each NeRV layer, a dedicated affine head maps all the weight tokens into modulation parameters. This strategy significantly expands the expressiveness of the weight tokens, as a compact latent space will reduce the complexity of the diffusion process in the generation stage.

**Channel-wise INR Parameterization.**   FastNeRV repeats the weight tokens and multiply them to the instance-agnostic INR base weights via dot product as the modulation. Skorokhodov et al. (2021); Yu et al. (2022) perform low-rank vector cross product to amplify the modulation matrix dimension from condensed weight latent. Inspired by Lin et al. (2021) that prunes a pretrained GAN generator by subsetting its kernels, we propose to directly set affined instance-specific weight tokens to be the convolutional kernels at a certain group of INR channels. Other parameters $\theta_s$ are shared among all training data and are learnable during training. All kernel values are normalized along all dimensions except the output channels, following the demodulation in Karras et al. (2019). In this way, the generated weight tokens are directly involved in decoding with maximal degrees of freedom. This also enables smooth parameter interpolation between our INR decoders.

**Convolutional Discriminator.**   To generate realistic videos we incorporate adversarial training (Goodfellow et al., 2020). We choose a convolutional discriminator (Karras et al., 2019) over a transformer-based one, as we observe that the latter introduces flickering artifacts across frames.

**Training Objectives.**   We train NeRV-VAE with the reconstruction objective. With an additional perceptual loss (Zhang et al., 2018) $\mathcal{L}_{\text{LPIPS}}$ and the adversarial loss $\mathcal{L}_{\text{GAN}}$, it is optimized via

$$\mathcal{L}_{\text{VAE}}(\mathcal{E}, \theta_s) = \|x - \tilde{x}\|^2 + \mathcal{L}_{\text{LPIPS}}(x, \tilde{x}) + \mathcal{L}_{\text{GAN}}(x, \tilde{x}) + D_{\text{KL}}(\mathcal{N}(0, 1), \tilde{\theta}). \tag{1}$$

### 3.2 Generative NeRV Decoder

The encoded weight tokens are formed into a video INR $\mathcal{D}_\theta(\cdot)$ that decodes to reconstruct the video. NeRV (Chen et al., 2021a) is a convolutional video INR that takes time index $t$ as the input query and yields a whole frame at each forwarding. We construct our implicit decoder based on it while introducing several upgrades to enhance its capacity for generative purposes.

**Spatiotemporal Embedding Input.**   Time-query video INRs upsamples from $\mathbb{R}^{T \times D \times 1 \times 1}$ to $\mathbb{R}^{T \times 3 \times H \times W}$, where no spatial dimension is input. With this structure, we observe distinct movement in the reconstruction, however the spatial content lacks clarity. To balance between the appearance and motion quality, we expand the input time embedding to 3D spatiotemporal, while time remains the sole query axis. Specifically, we sample a 3D positional embedding and reshape it to $\mathbb{R}^{T \times 3D \times h \times w}$. This spatiotemporal input supplements geometric prior and avoids the leading FC layer in vanilla NeRV that were designed for transforming 1D time embedding input. We observe that full convolutions fit optimally for generative quality with our multi-head affine modulation.

**Scaling up Blocks.**   Benefited from the reused weight modulation with multi-head affine mappings, we are able to largely scale up our NeRV decoder without extra weight tokens. We expand the upsampling layers to blocks, each performing one-level ($2\times$) upsampling with additional convolutions that don't change the shape. Compared to the assorted upsampling scales in limited layers in vanilla NeRV, this periodic upsampling structure evenly distributes the information from low to high resolutions, and cooperates well with our multi-head affine modulation. We also double the hidden dimensions of the layers in the last block following Karras et al. (2020) so that more native high-resolution information can be processed with sufficient capacity.

**Upsampling Algorithm.**   While vanilla NeRV has tested that pixelshuffle results in the best reconstruction performance with similar amount of parameters, we again compare different upsampling algorithms for our NeRV decoder. We find that transposed convolutions achieve non-negligible better generation quality to pixelshuffle with merely a quarter of parameters and computations. Therefore we choose transposed convolutions for all the upsampling layers in our NeRV decoder.

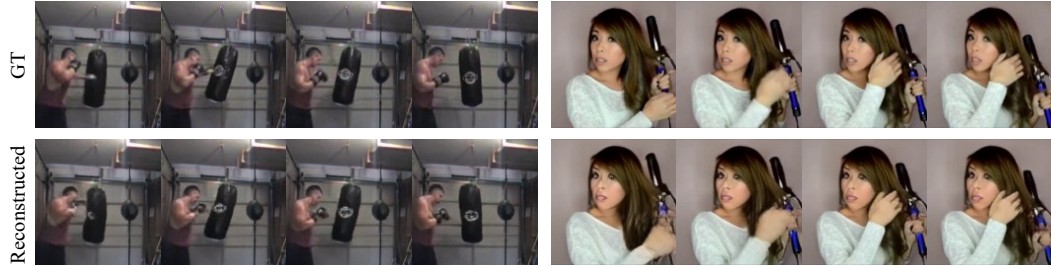

Figure 3: Video reconstruction of our NeRV-VAE on UCF (left) and K600 (right).

Table 1: Generative performance comparison on UCF and K600 at different resolutions and lengths.

(a) UCF 16 frames at $128^2$ resolution.

| Method | gFVD↓ |
|---|---|
| *Non-Implicit Models* | |
| TATS (Ge et al., 2022) | 332 |
| MAGVIT-AR (Yu et al., 2023a) | 265 |
| VideoFusion (Luo et al., 2023) | 173 |
| MAGVITv2-AR (Yu et al., 2024a) | 109 |
| LARP-L (Wang et al., 2025a) | 102 |
| *Implicit Models* | |
| DIGAN (Yu et al., 2022) | 465 |
| NeRV-Diffusion-S (Ours) | 184 |
| NeRV-Diffusion-B (Ours) | 133 |
| NeRV-Diffusion-L (Ours) | **97** |

(c) UCF 16 frames at $256^2$ resolution.

| Method | gFVD↓ |
|---|---|
| VIDM (Mei & Patel, 2023) | 263 |
| Latte (Ma et al., 2024) | 202 |
| OmniTokenizer (Wang et al., 2024a) | 191 |
| AR-Diffusion (Sun et al., 2025) | 186 |
| HPDM-M (Skorokhodov et al., 2024) | 143 |
| NeRV-Diffusion-L (Ours) | **140** |

(d) UCF 128 frames at $128^2$ resolution.

| Method | gFVD↓ |
|---|---|
| *Non-Implicit Models* | |
| Latte (Ma et al., 2024) | 1157 |
| PVDM (Yu et al., 2023b) | 505 |
| VIDM (Mei & Patel, 2023) | 426 |
| CoordTok (Jang et al., 2025) | 369 |
| *Implicit Models* | |
| DIGAN (Yu et al., 2022) | 1103 |
| NeRV-Diffusion-L (Ours) | **366** |

(b) K600 16 frames at $128^2$ resolution.

| Method | gFVD↓ |
|---|---|
| OmniTokenizer (Wang et al., 2024a) ($256^2$) | 33 |
| LARP-L (Wang et al., 2025a) | **17** |
| NeRV-Diffusion-L (Ours) | 22 |

**Side Connections.** With the increased depth of our NeRV decoder by upscaled blocks, we further append side connections to effectively collate all intermediate resolution information with minimal computation overhead. We investigate the residual and skip connections as in Karras et al. (2020). If the side connection needs additional layers, they are also modulated by the same set of our weight tokens thanks to our multi-head affine mappings and no extra trainable parameter is introduced. Residual connection fuses latent features at different scales before decoding to RGB and is experimented to yield clearer appearance and stabler motion.

### 3.3 IMPLICIT DIFFUSION

With visual data tokenized from pixel space to NeRV weight space by the implicit autoencoder described above, we perform diffusion process on these weight tokens by $\theta_t = \alpha_t \theta_0 + \sigma_t \epsilon$ and train a denoising network $\phi$ toward

$$\mathcal{L}_{\text{IDM}} = E_{\theta,\epsilon \sim \mathcal{N}(0,1),t}[\|\epsilon_0 - \epsilon(\epsilon_t, t)\|^2] \tag{2}$$

It is not trivial to model denoising process on neural weights. Previous diffusion models are designed for pixel data or their latent feature maps. Since NeRV weight tokens have no spatiotemporal structure, transformers are more suitable than U-Nets to process them, and temporal attention is unnecessary in our denoising network like those in traditional video diffusion models (Ma et al., 2024).

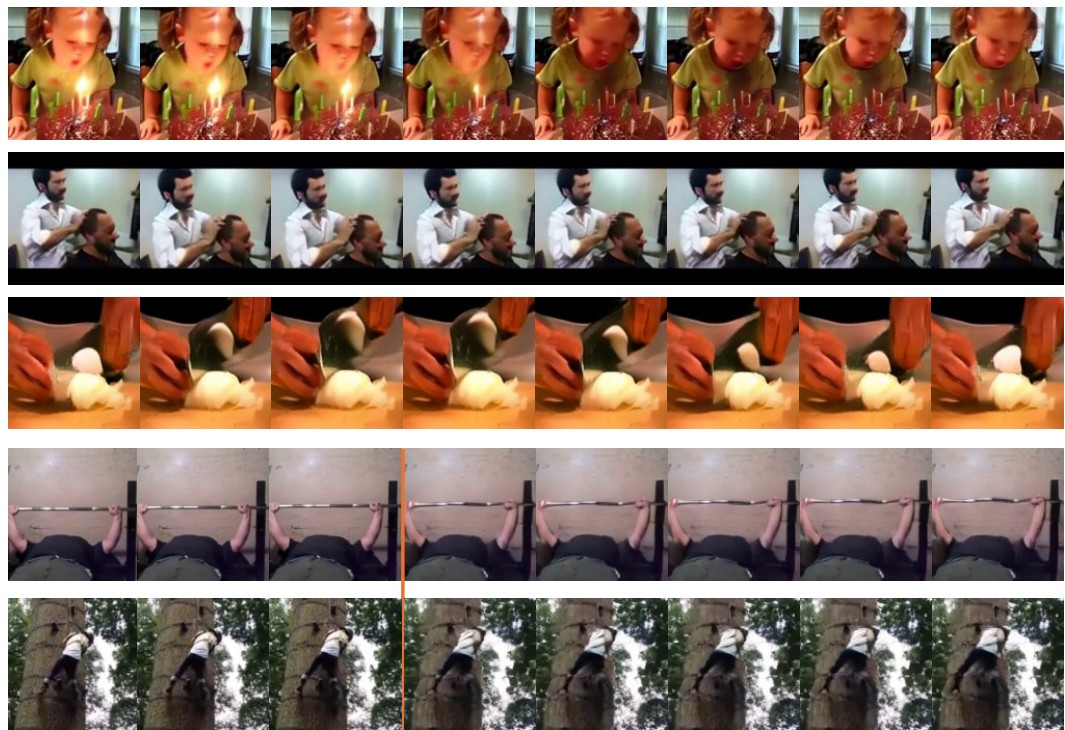

Figure 4: Class-conditioned video generation on UCF (rows 1-3) and video frame prediction on K600. (rows 4-5). Frames in front of the orange line are input conditions.

G.pt (Peebles et al., 2022) uses transformers to evolve neural network weights in a meta-learning fashion but not on noisy data. DiT (Peebles & Xie, 2023) tailors transformers for image diffusion and Zha et al. (2025) also use it to process 1D image tokens in diffusion. We explored these backbone options and DiT reaches the optimal performance with a straightforward architecture. Besides, we curate the training scheme as below to fill the gap when adapting DiT to the implicit space.

**Min-SNR-$\gamma$ Loss Weighting.** We observe that our implicit diffusion model converges slower on early denoising timesteps than late ones, i.e. it is tougher to learn to parse more noisy input. To address this issue and speed up its convergence, we adopt Min-SNR-$\gamma$ loss weighting (Hang et al., 2023) and apply the coefficient $w_t = \min\{\mathrm{SNR}(t), \gamma\}$ on the denoising loss, where $\mathrm{SNR}(t) = \frac{\alpha_t^2}{\sigma_t^2}$ reflects the signal-noise ratio at timestep $t$, and constant $\gamma$ controls the minimum of $w_t$. This loss weighting strategy prevent the diffusion training to focus too much on the low noise levels and descends evenly toward the denoising directions at all timesteps.

**Scheduled Sampling.** To further enhance the implicit denoising chain and tackle the exposure bias issue, we introduce scheduled sampling (Bengio et al., 2015) into our training scheme. It is initially proposed for auto-regressive models, and has been applied on diffusion models (Ning et al., 2023; Ren et al., 2024) to fill the training-inference gap brought by Teacher Forcing. During training, after the first forward round at step $t$, we randomly use the model predictions $\tilde{\theta_{t-1}} = \theta_\phi(\theta_t, t)$ as the new input and execute another forward pass, and calculate the total losses. It aligns the the training and inference modes, ensures low input disparity and minimizes error accumulation during sampling.

## 4 EXPERIMENTS

### 4.1 SETUPS

**Datasets.** We demonstrate NeRV-Diffusion on two video benchmarks: video generation on UCF-101 (Soomro et al., 2012) (UCF) and frame prediction on Kinetics-600 (Kay et al., 2017) (K600).

Table 2: Decoding (top) and generation (bottom) efficiency comparisons at 128 and 256 resolutions. All results are tested on a single NVIDIA A6000 GPU in `bfloat16` at batch size 1 and averaged over 100 runs. All generators have enabled CFG and iterate for their default sampling timesteps.

| Decoder | #Params | | #Tokens | | Latency↓ | | VRAM↓ | |
|---|---|---|---|---|---|---|---|---|
| | $128^2$ | $256^2$ | $128^2$ | $256^2$ | $128^2$ | $256^2$ | $128^2$ | $256^2$ |
| Latte (Ma et al., 2024) | - | 49M | - | - | - | 0.288s | - | 5.2G |
| CMD (Yu et al., 2024c) | 24M | 24M | 272 | 1040 | 0.129s | 1.030s | 1.2G | 4.1G |
| Open-MAGVIT2 (Luo et al., 2024) | 226M | - | 1024 | - | 0.127s | - | 45G | - |
| SD-VAE (Rombach et al., 2022) | 49M | 49M | 4096 | 16384 | 0.048s | 0.260s | 1.2G | 4.3G |
| SVD-VAE (Blattmann et al., 2023a) | 98M | 98M | 4096 | 16384 | 0.094s | 0.411s | 1.3G | 4.5G |
| NeRV-VAE-L (Ours) | 55M | 64.5M | 128 | 160 | 0.032s | 0.133s | 0.99G | 2.6G |

| Generator | #Params | #Tokens | | Steps | Latency↓ | | VRAM↓ | |
|---|---|---|---|---|---|---|---|---|
| | | $128^2$ | $256^2$ | | $128^2$ | $256^2$ | $128^2$ | $256^2$ |
| Latte (Ma et al., 2024) | 674M | - | 512 | 250 | - | 37s | - | 4.4G |
| LARP-L (Wang et al., 2025a) | 343M | 1024 | - | 1024 | 20s | - | 1.6G | - |
| OmniTokenizer (Wang et al., 2024a) | 650M | - | 1280 | 5120 | - | 139s | - | 4.5G |
| NeRV-Diffusion-L (Ours) | 467M | 128 | 160 | 250 | 6.8s | 8.2s | 1.8G | 2.1G |

We conduct experiments on 16 or 128 frames of $128^2$ and $256^2$ resolution. Note that for 128-frame experiments we subsample the raw videos with frame interval of 8 to train both stages, and only produce 128 in diffusion sampling with temporal interpolation of the input frame indices (§4.6.1). We use the train split of K600, and all videos from UCF, following prior work (Yu et al., 2023a).

**Implementations.** We realize our NeRV encoder with the backbone of Vision Transformer (Dosovitskiy et al., 2021) (ViT). We scale up our NeRV decoder to three configurations of progressive sizes: -Small (3.5M), -Base (14M) and -Large (55M). We use 128 weight tokens with 128 channels for all configurations. We ablate our key design options in §4.5. Detailed model and training configurations are provided in Appendix A.

**Metrics.** We measure Fréchet Video Distance (FVD) (Unterthiner et al., 2018) to evaluate the generation quality of NeRV-Diffusion. We calculate FVD on 2,048 sampled videos following prior work (Yu et al., 2022; 2023a; Wang et al., 2025a) for fair comparison. We adopt additional metrics for reconstruction and generation in Appendix D.2.

## 4.2 VIDEO RECONSTRUCTION

Visualized reconstruction output of our implicit tokenizer are displayed in Figure 3. NeRV-Diffusion achieves comparable performance to other non-implicit methods, with much more compact model and latent sizes. It also worth noting that NeRV-Diffusion features a small reconstruction-generation gap compared to other models, indicating our effective design of implicit video representations for generation purposes, and thus efficient usage of our latent space.

## 4.3 VIDEO GENERATION

**Class-Conditioned Video Generation.** We conduct class-conditioned video generation on UCF and present our visual results in Figure 4. We quantitatively compare NeRV-Diffusion with other models in Table 1a. NeRV-Diffusion outperforms previous INR-based generative methods as well as most recent non-implicit models of various mechanisms, including GAN, diffusion and autoregressive architectures. It is able to synthesize dynamic videos with diversity in both appearances and motions, ranging from detailed objects to complex scenes. More visualizations and qualitative comparisons are displayed in Appendix D.1.

**Video Frame Prediction.** Following Hong et al. (2022), we train our implicit diffusion model given the initial 5 frames to predict the rest 11 frames. We construct a sequence of the 5 given

Table 3: Ablation studies of the key design options in our NeRV-VAE, tested with NeRV-Diffusion-S and the best implicit denoiser configuration on UCF.

| Modulation | gFVD↓ |
|---|---|
| Repeat | 741 |
| FMM | 636 |
| Channel | **570** |

(a)

| Reuse | gFVD↓ |
|---|---|
| No reuse | 570 |
| Direct reuse | 562 |
| Multi-head affines | **283** |

(b)

| Spatial PE | gFVD↓ |
|---|---|
| $h = w = 1$ | 283 |
| $h = w = 4$ | 269 |
| $h = w = 8$ | **254** |
| $h = w = 16$ | 277 |

(c)

| Upsampling | gFVD↓ |
|---|---|
| PixelShuffle | 254 |
| Transposed Conv | **248** |
| Bilinear | 287 |

(d)

| Side Connection | gFVD↓ |
|---|---|
| Vanilla | 248 |
| Residual | **219** |
| Skips | 235 |

(e)

| Token Shape | gFVD↓ |
|---|---|
| $32 \times 128$ | 219 |
| $64 \times 128$ | 193 |
| $128 \times 128$ | **184** |
| $256 \times 128$ | 206 |

(f)

Table 4: Ablation studies of the key design options in our implicit diffusion model, tested with NeRV-Diffusion-S and the best NeRV-VAE configuration on UCF.

| Model | gFVD↓ |
|---|---|
| G.pt (Peebles et al., 2022) | 550 |
| DiT (Peebles & Xie, 2023) | **295** |
| Latte (Ma et al., 2024) | 342 |

(a)

| Configurations | gFVD↓ |
|---|---|
| Vanilla DiT | 295 |
| w/ Min-SNR-$\gamma$ | 238 |
| w/ Scheduled Sampling | 261 |
| w/ Both | **184** |

(b)

frames and 11 duplicate 5th frames and encode them as a video clip to the NeRV weight space. We double the input channels of the DiT input embedder to fuse the clean condition and noise. The quantitative results are listed in Table 1b and the visualizations are displayed in Figure 4. NeRV-Diffusion faithfully propagates the spatial content and movement flows to future frames.

**Scaling Up Video Resolution and Length** We scale up NeRV-Diffusion to $256^2$ resolution and 128 frames on UCF. We report the generation performance comparison in Tables 1c and 1d. In both cases our NeRV encoder remains the same size. NeRV-Diffusion outperforms recent SOTAs with high compactness and easy extensibility. Notably, NeRV-Diffusion features sublinear complexity overhead when scaling up video resolutions and lengths, which we address in the next section.

## 4.4 DECODING AND GENERATION EFFICIENCY

Conventional VAEs encode videos into frame-wise feature maps with fixed spatial and temporal downsampling factors, and their latent size thus increases quadratically w.r.t. to RGB resolutions and length. In contrast, NeRV-Diffusion appends an additional layer/block to the end to perform an extra upsampling to double the output resolution. Our generative NeRV also features smooth native time interpolation and can be trained with large frame intervals (§4.6.1). Therefore, our model parameters increase sublinearly, and the neural weight latent size also only needs to increase accordingly. Specifically, our NeRV decoder size increases by 17%, and we increase the neural latent token numbe by 25% for simple alignment of channel-wise parameterization.

We compare the inference speed and peak GPU memory of NeRV-Decoder and NeRV-Diffusion with other video decoders and generators, at 16 frames and both $128^2$ and $256^2$ resolutions. Results are listed in Table 2. Our models cost far less latency and VRAM footprint in both decoding and generation stages. It demonstrates the superior efficiency of our implicit framework, especially by obviating temporal attentions and reusing the same parameters to decode all frames with redundancy.

### 4.5 ABLATION STUDIES

We conduct ablation studies to assess the key components we propose in §3 and validate the optimal design options for generative objectives. The quantitative results are tested with NeRV-Diffusion-S configuration on UCF and are listed in Tables 3 and 4.

Table 3a indicates that in our NeRV-VAE, our channel-wise parameterization outperforms due to its maximal transparency to decode directly using the encoded weight tokens. In Table 3b, our multi-head affines significantly boost the capacity of NeRV by mapping the whole bottleneck weight tokens to different NeRV layers for reused modulation. Table 3c demonstrates that the spatiotemporal input embedding of shape $h = w = 8$ expands the input space with peak expressiveness, while smaller sizes lead to truncated space and bigger sizes result in fewer upsampling layers. We compare different upsampling operations in Table 3d, and find that transposed convolution surpasses pixelshuffle by much fewer parameters and computations without inflated channels. We further explore side connection types in Table 3e, and observe that residual connections fuse raw features at diverse scales without visible artifacts brought by skip connections when summing up multi-resolution RGB output. Finally we scale up our implicit latent space by increasing the number of tokens, as we meanwhile observe that the token dimension only makes slight impact on the output quality. 128 tokens reaches the peak performance and more tokens will lead to an over complex latent space for diffusion although the reconstruction error continues dropping.

For our implicit denoising network, we consider three backbone candidates in Table 4a. G.pt was designed for neural weight evolution, but not adapted to diffusion tasks. Latte was designed for video generation and incorporate with temporal attentions, which are not beneficial to our implicit generation as NeRV weight tokens lack spatiotemporal structure. Table 4b showcases that both Min-SNR-$\gamma$ loss weighting and scheduled sampling scheme effectively minimize the gap between implicit diffusion training and inference, by emphasizing the denoising model more on high-noise predictions and imperfect input.

### 4.6 PROPERTIES OF GENERATIVE NERV

#### 4.6.1 TIME INTERPOLATION AND EXTRAPOLATION

Benefited from the continuous frame index positional embedding, our generative NeRV features flexible time interpolation and extrapolation capability. In Figures A1 and A2, we interpolate the input time embeddings by a factor of $8\times$ to sample 128-frame videos with smooth and distinct motions. This property indicates that our generative NeRV efficiently encodes high-density information and understand the residual intrinsic of frame sequences. It enables compact representation of long videos and efficient training with fewer frames and large frame intervals.

#### 4.6.2 GENERATIVE NERV WEIGHT INTERPOLATION

Our generative NeRV also features smooth interpolation between two distinct videos by interpolating their instance parameters. Given two generative NeRVs' parameters $\theta_1$ and $\theta_2$, we perform linear interpolation $\lambda\theta_1 + (1 - \lambda)\theta_2$ between them. The visual outcomes are exhibited in Figure A3. Our model produces progressively interpretable results compared to DIGAN. We attribute this parametric continuity to not only the Gaussian distribution constraint of our weight latent, but also our simple yet effective linear bottleneck mapping and channel-wise parameterization.

## 5 CONCLUSION

We propose NeRV-Diffusion, a two-staged video synthesis model via NeRV weight generation. Our NeRV-VAE projects videos into a Gaussian weight latent space for tokenization, where our implicit diffusion model denoises to generate neural weights that render into videos. NeRV-Diffusion outperforms both INR-based and most recent non-implicit video generative models on multiple real-world video benchmarks, demonstrating promising scaling law. It also features smooth temporal and parametric interpolation properties. The outstanding performance of NeRV-Diffusion highlights the potential of a new holistic video synthesis paradigm with efficient representations.

## 6 Acknowledgments

This work was partially supported by NSF CAREER Award (#2238769) to AS. The authors acknowledge UMD's supercomputing resources made available for conducting this research. The U.S. Government is authorized to reproduce and distribute reprints for Governmental purposes notwithstanding any copyright annotation thereon. The views and conclusions contained herein are those of the authors and should not be interpreted as necessarily representing the official policies or endorsements, either expressed or implied, of NSF or the U.S. Government.

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

# A    IMPLEMENTATION DETAILS

## A.1    ADDITIONAL MODEL AND TRAINING DETAILS

We use a medium configuration of ViT with 18 blocks, 14 heads and 896 hidden dimensions for our NeRV encoder. We set the scale of KL divergence loss to $1 \times 10^{-5}$. It patchifies the RGB videos into $8 \times 8 \times 1$ patches along height, width and time dimensions. We use sinusoidal positional embedding for our generative NeRV's input time index, instead of the exponential embedding in vanilla NeRV. For residual connections we use bilinear to upsample the earlier feature maps before merging them into the main branch.

Our discriminator is adapted from a 3D StyleGAN with 5 blocks, 64 unit hidden dimensions and a channel multiplier of 2 for each block. Its learning rate is set to one fifth of the NeRV-VAE's, and it is updated every five iterations to stabilize the training. The scale of the GAN loss added to our NeRV-VAE is 1.

Our implicit diffusion transformer adopts DiT-L configuration with 24 layers, 16 heads and 1024 hidden dimensions. Its patch size follows the token shape as output by our NeRV encoder. Our implicit DiT is optimized for predicting the noise $\epsilon$ at each timestep, and thus also adjust the Min-SNR-$\gamma$ loss weighting accordingly. We employ CFG for class-conditioned sampling and the optimal guidance scale is 2.

Both our NeRV-VAE and implicit DiT train with L2 reconstruction loss. We use AdamW (Loshchilov, 2017) optimizer with a linear warmup learning rate schedule and cosine decay. Both learning rates are set to $1 \times 10^{-4}$. We train NeRV-VAE for 1M iterations and implicit DiT for 500K iterations at a small batch size of 32. We train each of them for 1 week on 8 NVIDIA A6000 GPUs.

## A.2    NERV DECODER ARCHITECTURE

We list the layer-wise architecture of our NeRV model in Table A1. "Feature Map" refers to the output feature map of each layer. "Modulation Weight" refers to the instance-specific weight latent to be assigned to each NeRV layers. $T$ is the number of frames.

We set the dimensions of the time, height and width positional embeddings all to 16. We start from sampling a spatiotemporal positional embedding of shape $[8, 8, T, 48]$. It is transposed to queries along the time axis $[T, 48, 8, 8]$, and then spatial convolutions are applied on it.

We use kernel size $k = 4$ for all upsampling transposed convolutions and $k = 3$ for all other convolutions that don't change the feature map shape. We set the base hidden dimensions $D = 128, 256, 512$ for NeRV-Diffusion-S, -B and -L configurations, respectively. `gelu` is used for activations in all blocks while `tanh` is used after the tailing toRGB layer.

# B    TIME INTERPOLATION

As discussed in §4.6.1, we train NeRV-Diffusion on UCF with an interval of 8 frames, and interpolate the input time embeddings by an $8\times$ factor to sample 128-frame videos. The results are presented in Figures A1 and A2.

**Quantitative Verification.**    We measure long video semantic consistency in terms of object identity and action logic preservation. Specifically, we leverage the Subject Consistency (SC) and Background Consistency (BC) metrics in VBench, which extract the subject and background features via pre-trained DINO (Caron et al., 2021), and calculate their pairwise similarities across all frames for appearance preservation. Inspired by it, we further employ C3D, a pre-trained action recognizer network, and feed it with all sub-clips from the long videos to extract action features. We calculate pairwise similarities of the action features across all windows to measure the action logic preservation. These metrics don't only rely on consecutive distance but average along the whole video, accurately reflecting the drifting issues for long video generation. The results are listed in Table A2.

Table A1: Detailed architecture of our generative NeRV Decoder and modulated weight shape. Batch size is omitted.

| Layer | Feature Map Shape | Modulation Weight Shape |
|---|---|---|
| Input Init | $[8, 8, T, 48]$ | - |
| Reshape | $[T, 48, 8, 8]$ | - |
| Conv | $[T, D, 8, 8]$ | $[48, D, 3, 3]$ or $[24, D/2, 3, 3]$ |
| Transposed Conv | $[T, D, 16, 16]$ | $[64, 64, 4, 4]$ |
| Conv | $[T, D, 16, 16]$ | $[64, 64, 3, 3]$ |
| Conv | $[T, D, 16, 16]$ | $[64, 64, 3, 3]$ |
| Transposed Conv | $[T, D, 32, 32]$ | $[64, 64, 4, 4]$ |
| Conv | $[T, D, 32, 32]$ | $[64, 64, 3, 3]$ |
| Conv | $[T, D, 32, 32]$ | $[64, 64, 3, 3]$ |
| Transposed Conv | $[T, D, 64, 64]$ | $[64, 64, 4, 4]$ |
| Conv | $[T, D, 64, 64]$ | $[64, 64, 3, 3]$ |
| Conv | $[T, 2D, 64, 64]$ | $[128, 64, 3, 3]$ |
| Transposed Conv | $[T, 2D, 128, 128]$ | $[64, 64, 4, 4]$ |
| Conv | $[T, 2D, 128, 128]$ | $[64, 64, 3, 3]$ |
| toRGB | $[T, 3, 128, 128]$ | $[D, 3, 3, 3]$ |
| Reshape to Output | $[T, 128, 128, 3]$ | - |

Table A2: Quantitative measurement of time interpolation semantic consistency.

| Long Consistency | VBench↑ | | Action Sim.↑ |
|---|---|---|---|
| | SC | BC | |
| Ground Truth | 0.931 | 0.956 | 0.912 |
| Ours | 0.919 | 0.942 | 0.901 |

## C  INR WEIGHT INTERPOLATION

We further illustrate our generative NeRV's superiority in INR weight interpolation. DIGAN (Yu et al., 2022) proposes to interpolate between the latent noise vectors. When being interpolated between the whole weights, their video INR presents non-continuous transitions as shown in Figure A3 (top). This is because 1) their latent vectors are decoded from Gaussian noise with a complex non-linear mapping network; 2) their INR weights are modulated with low-rank cross product, termed as Factorized Matrix Multiplication (FMM) in Skorokhodov et al. (2021)), of the latent vectors, which break the arithmetic property. In contrast, our weight latent is directly used for modulation with a single linear affine layer from the KL bottleneck, and is directly assigned as NeRV parameters with minimal transforms. Our generative NeRV presents smooth interpolation effect as shown in Figure A3 (bottom). This property also opens up the potential of general direct manipulations on the tokenized NeRVs in a compositional manner with our NeRV-VAE.

## D  ADDITIONAL EXPERIMENTS AND RESULTS

### D.1  VIDEO GENERATION

We provide more generated samples of NeRV-Diffusion on UCF in Figure A4.

**Qualitative Comparisons**  We present additional visual comparisons with LARP (Wang et al., 2025a) on UCF in Figures A5 and A6. We sample both models with the same class label input. Compared to LARP, NeRV-Diffusion also constructs holistic video representations but meanwhile still maintain the spatiotemporal integrity via the query time indices and spatial input embeddings, and thus produces more structural videos with less morphing or tearing.

## D.2 Additional Quantitative Metrics.

We extend our reconstruction metrics to PNSR, SSIM and LPIPS between the output and input videos, and extend our generation metrics to cross-frame LPIPS (fLPIPS) and C3D (Tran et al., 2015) / I3D (Carreira & Zisserman, 2017) -based Inception Score (Salimans et al., 2016) (IS) on the output videos. Due to the lack of pair-wise supervision in generation, we also evaluate generators on the non-text subsets of VBench Huang et al. (2024), including Subject Consistency (SC), Background Consistency (BC), Temporal Flickering (TF), Motion Smoothness (MS), Dynamic Degree (DD), Aesthetic Quality (AQ) and Imaging Quality (IQ). For comparing methods, we use their pretrained checkpoints for inference.

The results of reconstruction and generation on UCF and K600 are listed in Table A3. It worth noting that the ultimate purpose of NeRV-VAE (and the comparing tokenizers) is to encode RGB videos into smooth latent for high-quality video generation, and we optimize it for best generative quality instead of solely reconstruction faithfulness. Our models achieve SOTA generation performance on UCF across all dimensions, and reach comparable SOTA performance on K600 frame prediction.

Table A3: Additional reconstruction and generation metrics on UCF and K600.

(a) Additional reconstruction on UCF.

| Reconstruction | PNSR↑ | SSIM↑ | LPIPS↓ | rFVD↓ |
|---|---|---|---|---|
| TATS (Ge et al., 2022) | 22.43 | 0.765 | 0.108 | 162 |
| Open-MAGVIT2 (Luo et al., 2024) | 25.84 | 0.862 | 0.045 | 16 |
| LARP-L (Wang et al., 2025a) | 27.87 | 0.891 | 0.038 | 20 |
| NeRV-VAE-L | 26.63 | 0.879 | 0.043 | 22 |

(b) Additional reconstruction metrics on K600.

| Reconstruction | PNSR↑ | SSIM↑ | LPIPS↓ | rFVD↓ |
|---|---|---|---|---|
| LARP-L (Wang et al., 2025a) | 28.22 | 0.867 | 0.035 | 11 |
| NeRV-VAE-L | 26.45 | 0.823 | 0.044 | 19 |

(c) Additional generation metrics on UCF.

| Generation | fLPIPS↕ | IS↑ | VBench↑ | | | | | | | gFVD↓ |
|---|---|---|---|---|---|---|---|---|---|---|
| | | | SC | BC | TF | MS | DD | AQ | IQ | |
| Ground Truth | 0.031 | 86.24 | 0.954 | 0.977 | 0.981 | 0.988 | 0.282 | 0.410 | 0.443 | 0 |
| TATS (Ge et al., 2022) | 0.048 | 68.79 | 0.910 | 0.958 | 0.978 | 0.980 | 0.314 | 0.348 | 0.437 | 332 |
| VIDM (Mei & Patel, 2023) | - | 64.17 | - | - | - | - | - | - | - | 263 |
| VideoDiffusion (Luo et al., 2023) | - | 80.03 | - | - | - | - | - | - | - | 173 |
| LARP-L (Wang et al., 2025a) | 0.025 | 68.79 | 0.955 | 0.977 | 0.985 | 0.991 | 0.218 | 0.398 | 0.393 | 102 |
| NeRV-Diffusion-L | 0.028 | 82.17 | 0.958 | 0.978 | 0.983 | 0.989 | 0.262 | 0.392 | 0.433 | 97 |

(d) Additional generation metrics on K600.

| Generation | fLPIPS↕ | IS↑ | VBench↑ | | | | | | | gFVD↓ |
|---|---|---|---|---|---|---|---|---|---|---|
| | | | SC | BC | TF | MS | DD | AQ | IQ | |
| Ground Truth | 0.031 | 31.20 | 0.950 | 0.945 | 0.978 | 0.988 | 0.266 | 0.406 | 0.461 | 0 |
| LARP-L (Wang et al., 2025a) | 0.029 | 23.51 | 0.933 | 0.973 | 0.981 | 0.989 | 0.292 | 0.337 | 0.293 | 17 |
| NeRV-Diffusion-L | 0.034 | 27.09 | 0.928 | 0.967 | 0.977 | 0.979 | 0.354 | 0.392 | 0.432 | 22 |

## D.3 Downstream Tasks with Controllability

Our implicit diffusion model follows standard diffusion framework but only switches to neural weight latent, and thus can be trained with versatile condition types. Additional condition input can be integrated with standard cross attention so that different modalities other than videos or images are also supported. We extend NeRV-Diffusion to downstream tasks including image-to-video

generation and unconditional video generation. To showcase the support of granular controllability of our neural weight latent, we also include an edge-to-video generation experiment. We present qualitative results in Figure A7. These experiments demonstrate the native extensibility of NeRV-Diffusion spanning from none to fine-grained condition controls with high quality.

# E    Additional Discussions

## E.1    Dataset Details

UCF dataset contains over 13K real-world videos sourced from YouTube of average 180 frames, totally about 27 hours. K600 dataset contains about 500K videos of average 250 frames, summing up to about 57 days in length.

## E.2    Two-stage Generative Framework and Reconstruction-Generation Trade-off

Two-stage tokenizer-generator frameworks are widely adopted for their stability and efficiency compared to single-stage generation models, especially when scaled up to large-scale models and data. All generative tokenizers need to balance between the latent compactness and expressive richness, such as the KL constraint in VAE andthe codebook alignment for VQGANs.

The gap between gFVD and rFVD is controlled by the KL loss scale. A high KL loss scale will harm rFVD but reduce the gap between gFVD and rFVD, while a low KL loss scale will improve rFVD but enlarge the gap between gFVD and rFVD. We ablate its impact on NeRV-Diffusion-S configuration in Table A4.

Table A4: Ablation on KL loss scales on NeRV-Diffusion-S configuration.

| KL Loss Scale | $1 \times 10^{-6}$ | $5 \times 10^{-6}$ | $1 \times 10^{-5}$ | $5 \times 10^{-5}$ |
|---|---|---|---|---|
| **rFVD** | 73 | 82 | 85 | 107 |
| **gFVD** | 198 | 186 | 184 | 202 |

Moreover, we conduct marginal tests and normality plots on our encoded neural weight latent, and compare it to LDM/SD's VAE latent. The statistics and visualizations are presented in Table A5 and Figure A8.

Table A5: Latent Gaussianity statistics.

| Gaussianity | mean | std | skew | kurtosis |
|---|---|---|---|---|
| LDM/SD-VAE | $-0.0165$ | 5.2505 | $-0.4756$ | 0.1693 |
| NeRV-VAE-L | $-0.0036$ | 0.9979 | $-0.0001$ | 0.2568 |

## E.3    Granularity of Neural Network Weights

Our neural latent forms up the convolution weights of NeRV decoder, taking in unified spatiotemporal input embeddings to render pixel frames. While it is holistic and different from traditional framewise feature maps, it still performs convolutions, i.e. matrix dot productions over the spatiotemporal input grid (in the opposite positions), and thus still maintains spatiotemporal information. This is a major difference of our implicit representations between discrete holistic video latent such as LARP Wang et al. (2025a).

We design a probing experiment to validate the spatiotemporal association between our neural weight latent and pixel frames. We crop RGB videos into random clips with random spatial and temporal ranges, and encode them using various video encoders. For each video encoder, we perform K-Means on its encoded latent, with the raw video sources as the class labels. We calculate the

purity and normalized mutual information (NMI) between the latent clusters and their real labels, i.e. which raw video the corresponding clip is cropped from. Table A6 shows that our neural weight latent has spatiotemporal relations with pixel operations, indicating its potential to facilitate granular editing.

Table A6: Time interpolation semantic consistency.

| Granularity | Purity↑ | NMI↑ |
|---|---|---|
| SD-VAE | 0.484 | 0.753 |
| LARP | 0.285 | 0.591 |
| Ours | 0.366 | 0.656 |

### E.4 COMPARISON WITH OPEN-WORLD LARGE-SCALE VIDEO TOKENIZERS.

LTX (HaCohen et al., 2024) and WAN (Wan et al., 2025) are commercial foundation models that are trained on open-world large-scale data with considerable resources and extensive time. Though, we list the quantitative comparisons in Table A7. We also include the VAE of Stable Diffusion (SD) and Stable Video Diffusion (SVD), which are also large-scale foundation models at earlier stages.

Table A7: Comparison of reconstruction performance on UCF with foundation video tokenizers.

| Reconstruction (UCF) | PNSR↑ | SSIM↑ | LPIPS↓ | rFVD↓ |
|---|---|---|---|---|
| SD-VAE | 24.79 | 0.783 | 0.068 | 63 |
| SVD-VAE | 27.24 | 0.856 | 0.058 | 39 |
| LTX-VAE | 31.29 | 0.909 | 0.061 | 32 |
| WAN-VAE | 31.14 | 0.935 | 0.022 | 7 |
| NeRV-VAE-L | 26.63 | 0.879 | 0.043 | 22 |

### E.5 NEURAL WEIGHT LATENT DIMENSION ABLATION

We ablate different neural latent token dimensions with NeRV-Diffusion-S configuration in Table A8. It results in a slight performance drop when being reduced, but not as significant as token numbers shown in Table 3f. We attribute this to that our neural weight latent is holistic representations unlike traditional frame-wise feature maps, and the token dimension doesn't correspond to the RGB color channel.

Table A8: Ablations on neural weight latent dimensions.

| Token Shape | $128 \times 32$ | $128 \times 64$ | $128 \times 96$ | $128 \times 128$ |
|---|---|---|---|---|
| rFVD↓ | 114 | 99 | 92 | 85 |
| gFVD↓ | 230 | 212 | 197 | 184 |

### E.6 ENCODING AND DECODING EFFICIENCY ON 720P

We create random data and test the encoding/decoding latency on a $1280 \times 720$ video. The encoding takes approximately 1.9s, and the decoding takes approximately 1.8s, both on a single NVIDIA A6000 GPU.

### E.7 PERFORMANCE CEILING OVER EFFICIENCY

So far we haven't noticed such a ceiling as NeRV-Diffusion surpasses previous SOTA methods of both diffusion and autoregressive models. Meanwhile, the outstanding efficiency of NeRV-Diffusion

also indicates its extra potential when scaling up. If performance is prioritized over efficiency, NeRV-Diffusion can also further scale up to comparable efficiency for extra performance boost.

## F    REPRODUCIBILITY STATEMENT

Our code and trained checkpoints will be made publicly available upon publication. We have discussed our complete implementation details in §4.1 and Appendix A, including the model configuration and training recipe.

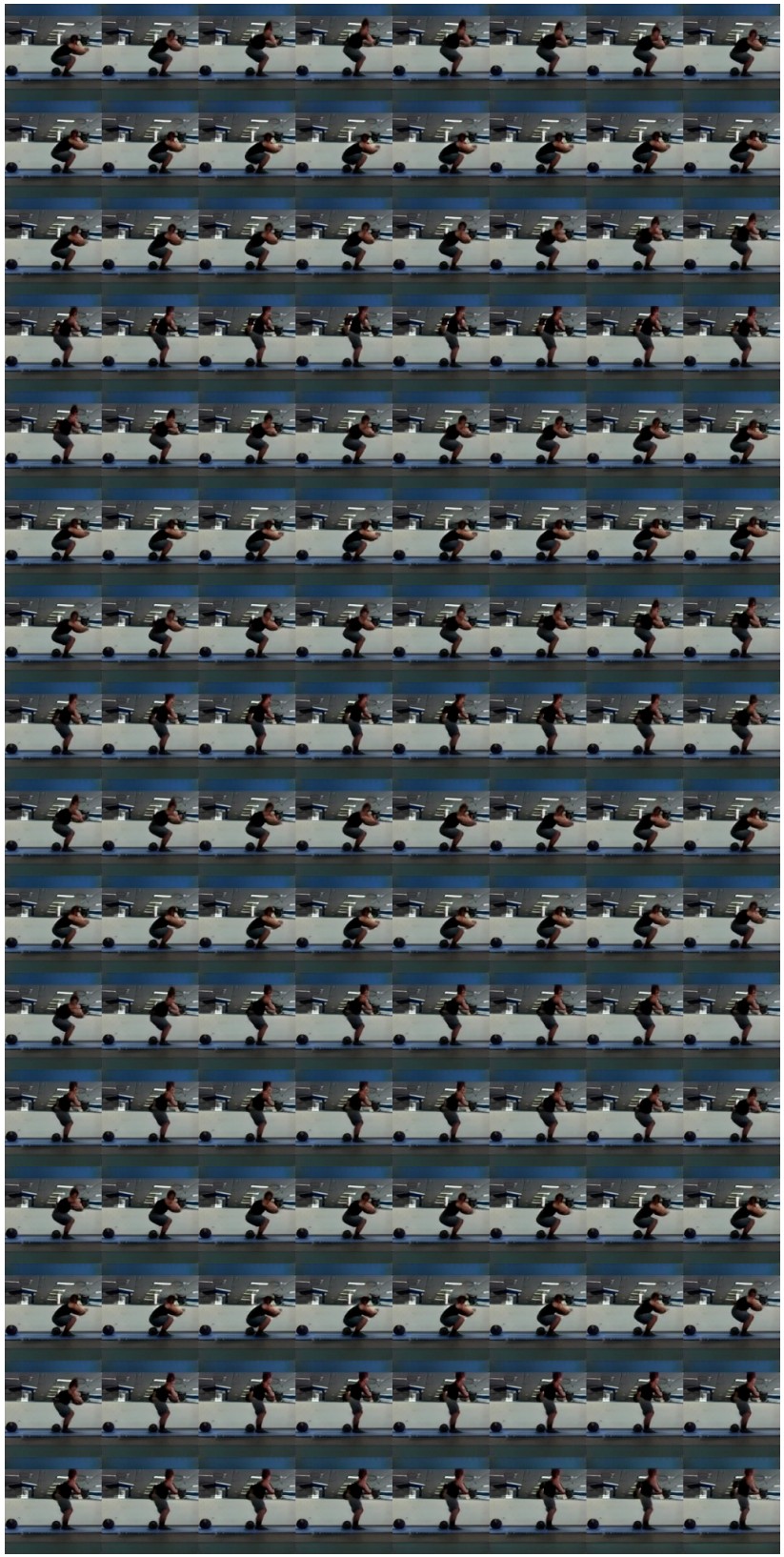

Figure A1: 128-frame video generation results on UCF. NeRV-VAE and NeRV-Diffusion only see one in every 8 frames in training, and is able to temporally generalize to unseen time stamps.

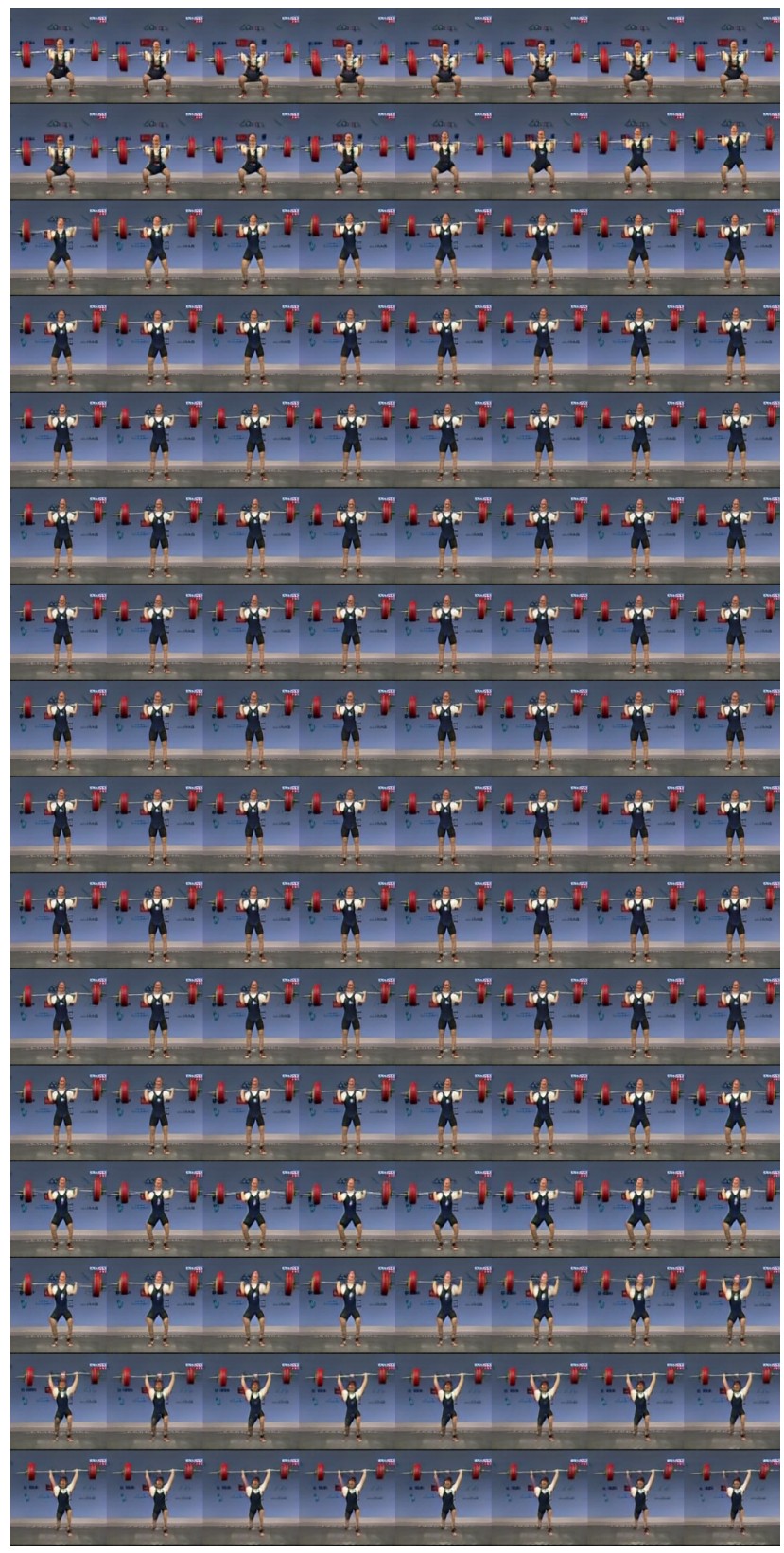

Figure A2: 128-frame video generation results on UCF. NeRV-VAE and NeRV-Diffusion only see one in every 8 frames in training, and is able to temporally generalize to unseen time stamps.

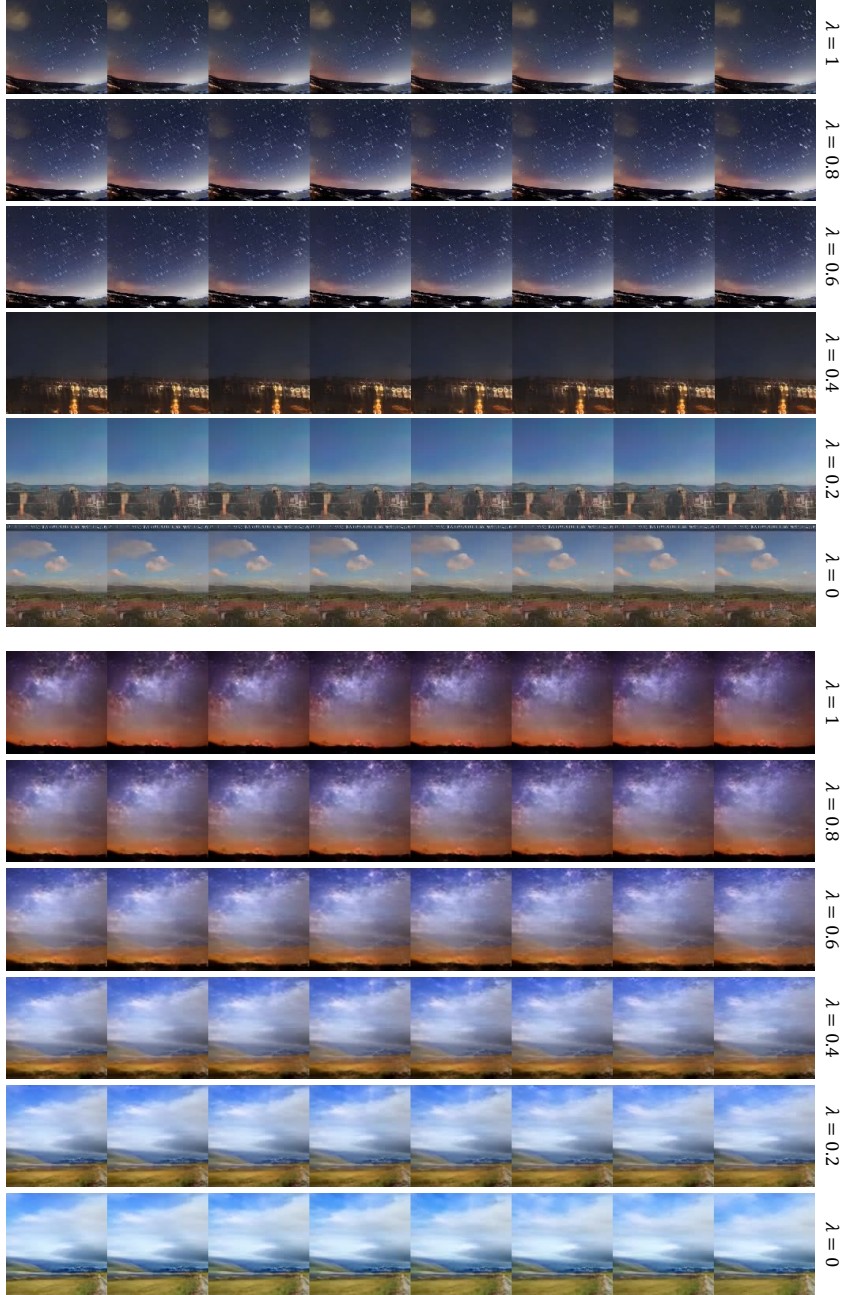

Figure A3: Interpolation of the whole parameters of the video INRs in DIGAN (top) and our generative NeRVs (bottom).

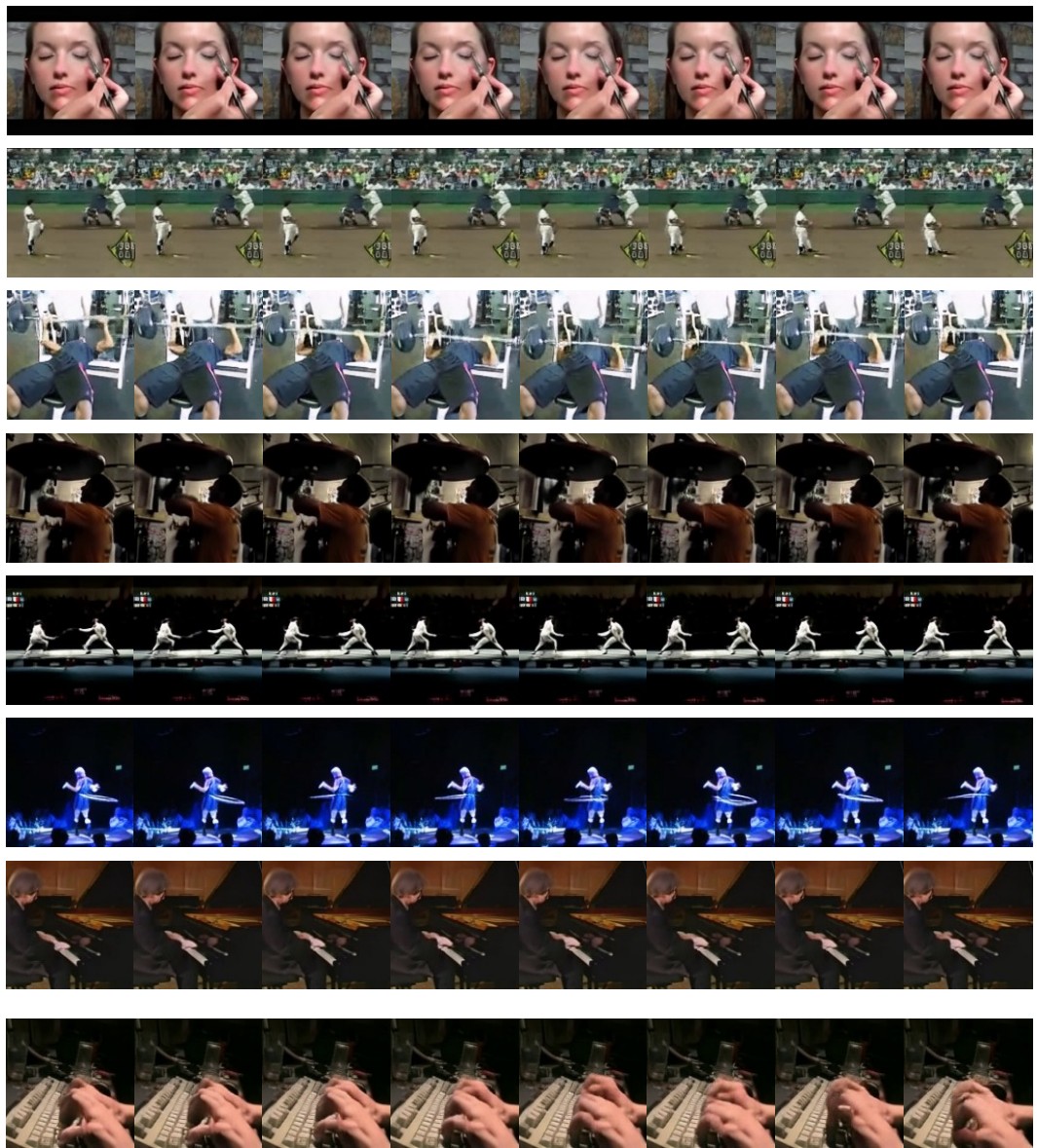

Figure A4: Additional class-conditioned generation samples on UCF.

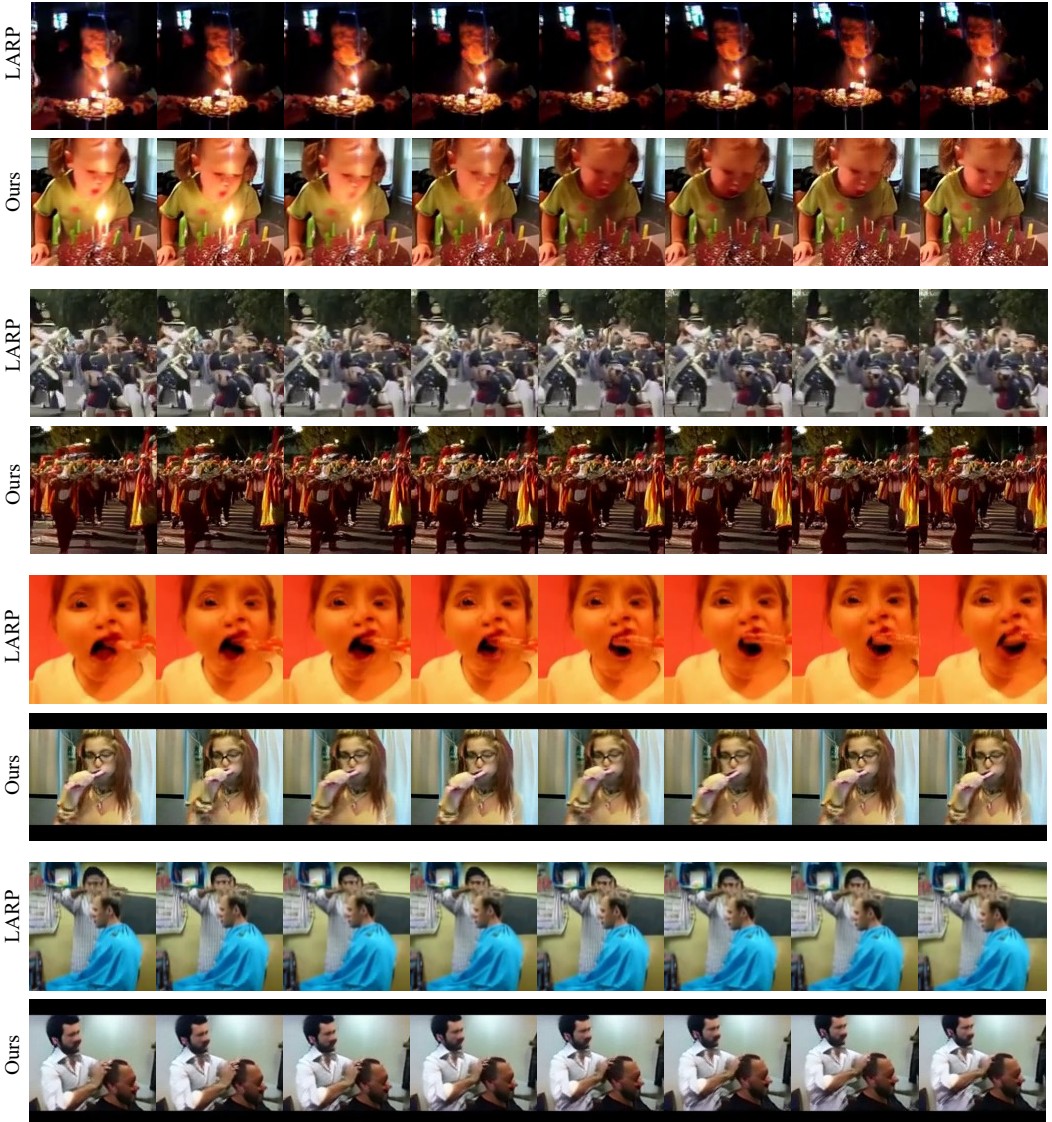

Figure A5: Qualitative comparisons with LARP on UCF class-conditioned generation.

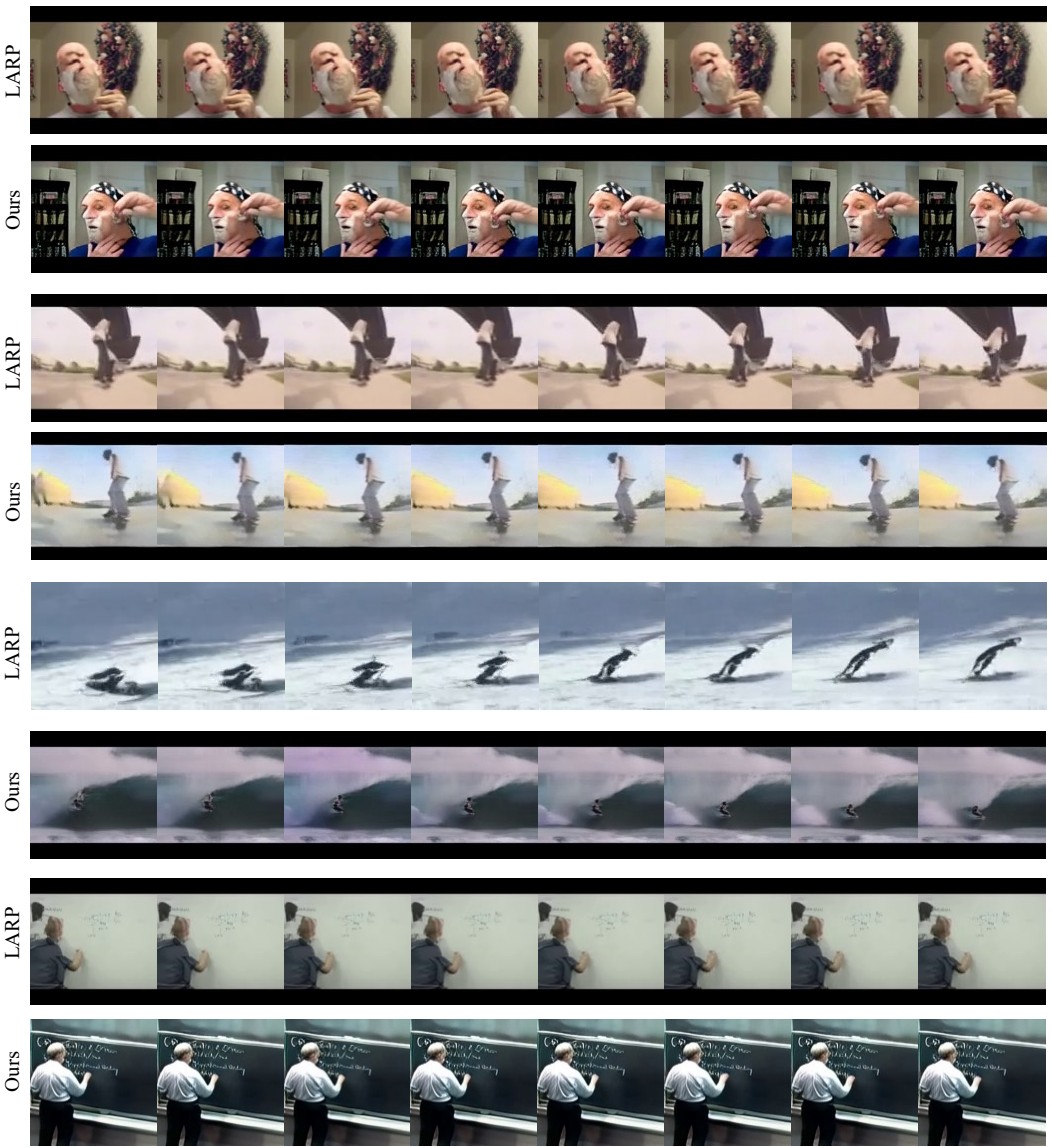

Figure A6: Qualitative comparisons with LARP on UCF class-conditioned generation.

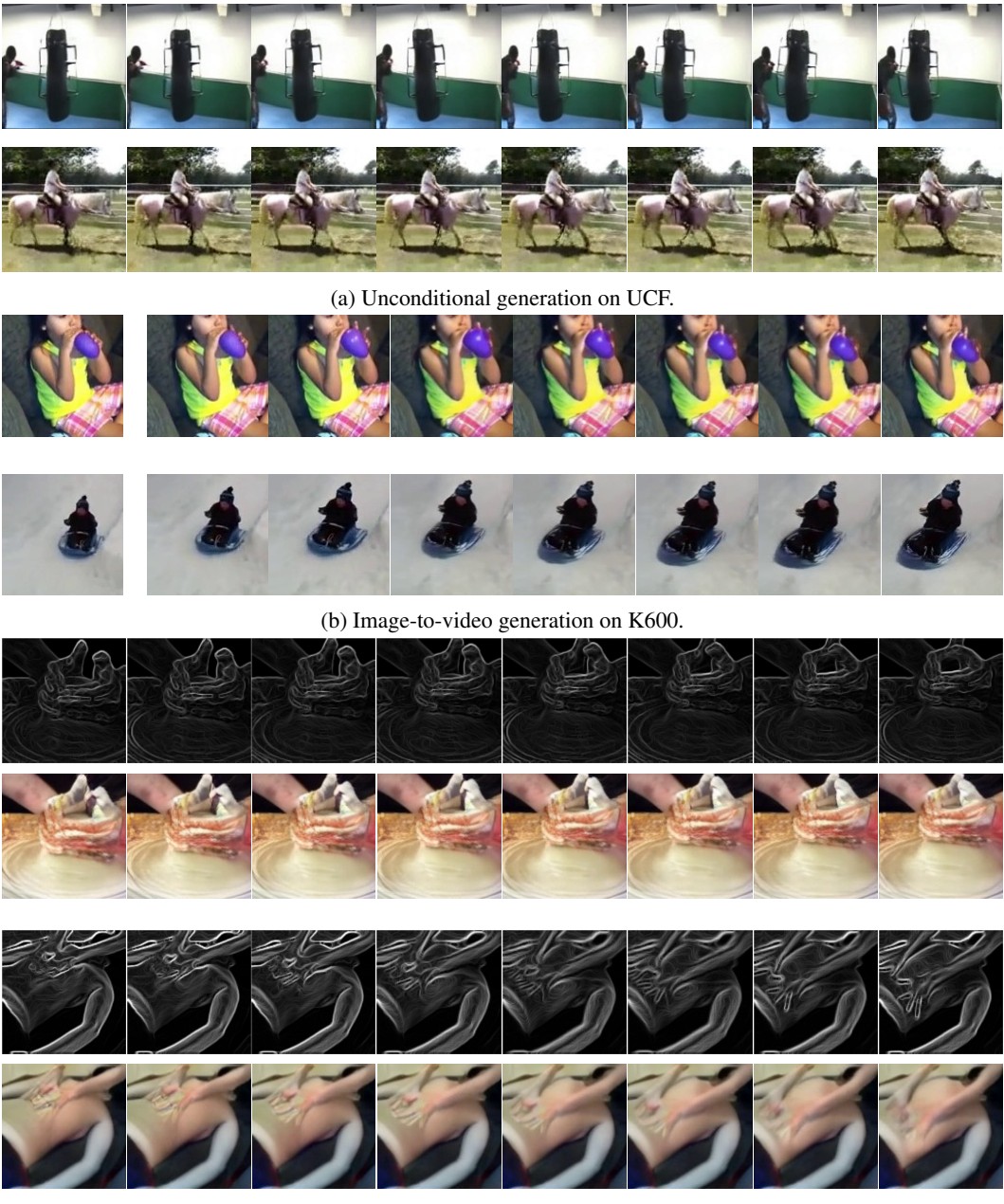

(a) Unconditional generation on UCF.

(b) Image-to-video generation on K600.

(c) Edge-conditioned video generation on K600.

Figure A7: Versatile condition types of video generation supported by NeRV-Diffusion.

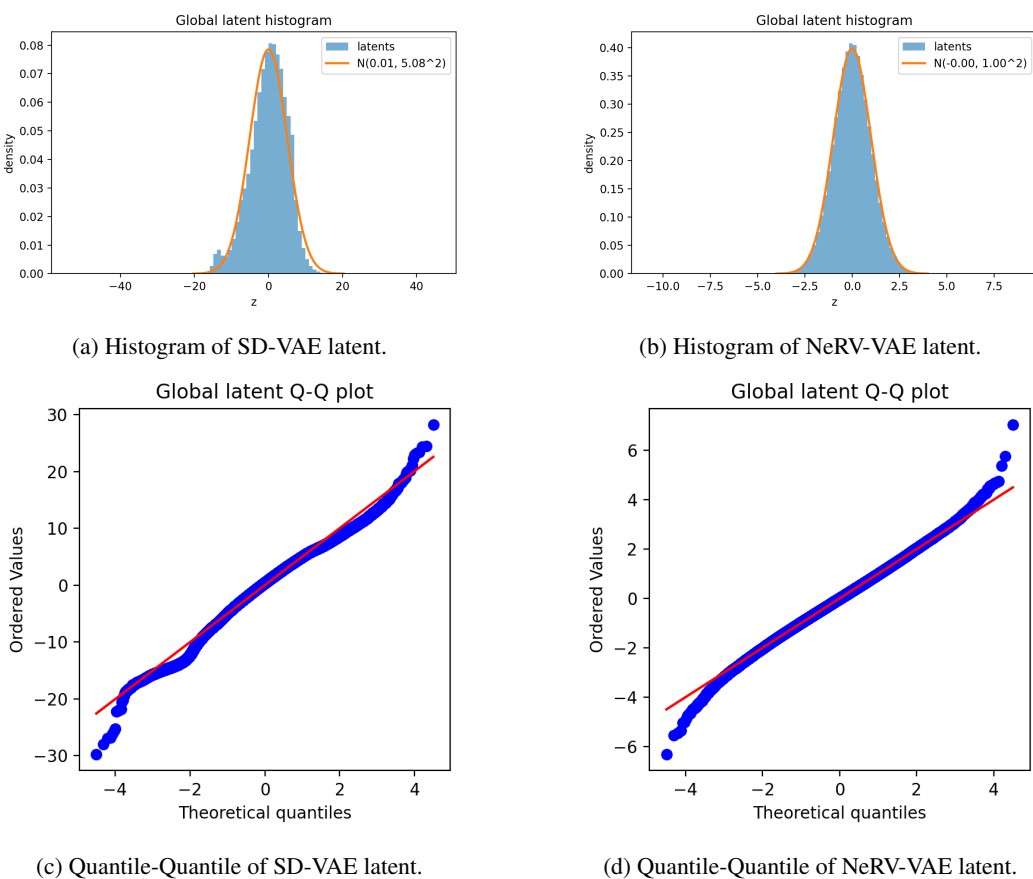

(a) Histogram of SD-VAE latent.

(b) Histogram of NeRV-VAE latent.

(c) Quantile-Quantile of SD-VAE latent.

(d) Quantile-Quantile of NeRV-VAE latent.

Figure A8: Gaussianity verifications of the latent encoded by NeRV-VAE and SD-VAE.

