# OpenReview forum: "NeRV-Diffusion: Diffuse Implicit Neural Representation for Video Synthesis"
_ICLR.cc/2026/Conference — ICLR 2026 Poster_

### Official Review · Reviewer_WpVC · 2025-10-28

**Soundness:** 3
**Presentation:** 2
**Contribution:** 4
**Rating:** 6
**Confidence:** 4

**Summary:**

This paper proposes **NeRV-Diffusion**, a novel implicit video diffusion model that synthesizes videos by generating the weights of an INR. The framework consists of two stages:

- **Tokenization stage**: A hypernetwork-based encoder compresses raw videos into a compact set of neural weights (i.e., INR parameters). These weights directly instantiate a NeRV  decoder, which reconstructs the video given frame indices as input.
- **Generation stage**: An DiT is trained in this implicit weight space to denoise from Gaussian noise and generate new INR weights, thereby synthesizing novel videos.

Experiments show that NeRV-Diffusion matches or surpasses current state-of-the-art non-implicit video generation models (e.g., MAGVITv2-AR, LARP) on UCF-101 and Kinetics-600, while using significantly smaller model and latent sizes. It also supports high-quality **temporal interpolation** and **INR weight interpolation**.

**Strengths:**

- **Highly innovative**: First work to combine diffusion models with video INR weight generation, establishing a new paradigm of “holistic video synthesis via neural weights.”
- **Technically solid**: Key architectural improvements to NeRV (e.g., channel-wise parameterization, multi-head affine modulation) are well-motivated and validated via ablation studies.
- **Comprehensive experiments**: Evaluated on UCF-101 and Kinetics-600 against GANs, autoregressive (AR), and diffusion-based models, with multi-dimensional assessments including reconstruction, generation, frame prediction, and interpolation.

**Weaknesses:**

- **Lack of quantitative efficiency analysis**: Although “fast decoding” is claimed, the paper does not report inference speed (e.g., FPS), memory footprint, or direct comparisons with LDM/VQ-VAE.
- **Scalability and resolution limitations**: All experiments are conducted only on 128×128×16 videos. Performance at higher resolutions (e.g., 256×256) or longer sequences (>16 frames) is not evaluated.
- **Limited analysis of long-video interpolation**: While long videos are generated via time interpolation, it remains unclear whether semantic consistency (e.g., object identity, action logic) is preserved.
- **Insufficient comparison with recent tokenization methods**, such as LTX Video VAE or WAN VAE.

**Questions:**

**On Efficiency**:
- Could you provide quantitative comparisons between NeRV-Diffusion and MAGVITv2 or Latte in terms of **inference latency** and **GPU memory usage**?
- When generating 128-frame videos, does the per-frame forward pass of the NeRV decoder become a bottleneck?
- What is the encoding/decoding efficiency when handling 720p video reconstruction and generation?
- How does the **training efficiency** (e.g., GPU-hours, convergence speed) compare to standard video VAEs?


**On Model Architecture**:
- What exactly does **Multi-head Affine Modulation** do? How does it enable the reconstruction of a full video from a single set of latent tokens?

**On Latent Dimensionality**:
- Each token has a dimension of 128, which seems large compared to recent DiT-based models (e.g., Sora/WAN), whose VAEs use only 16 or 48 dimensions. Why not adopt a lower latent dimension?
- With only **128 tokens**, how is high-fidelity reconstruction still achievable? What factors determine this efficiency?
- Could you report **PSNR, SSIM, and LPIPS** scores for reconstruction?
- Even if training data differs, can you compare **compression rate** and reconstruction quality against LTX Video VAE or WAN 2.2 VAE?

**On Performance Bottlenecks**:
- Why was the token count fixed at 128? What are the reconstruction and generation results with 512 or 1024 tokens?
- If one prioritizes **maximum performance over efficiency**, does this method have a relatively low ceiling compared to conventional latent diffusion approaches?

---

> ### Author Response · Authors · 2025-11-26
>
> We appreciate your acknowledgment of the high innovativeness, technical solidness and comprehensive experiments of our work.
> We address your concerns and questions as below:
>
> **W1: Decoding efficiency comparison.**
>
> We compare the inference speed and peak GPU memory of NeRV-Decoder with other models, including research projects that are trained on similar UCF/Kinetics benchmarks and large-scale foundation models that are trained on open-world data.
> All results are tested on a single NVIDIA A6000 GPU in bfloat16.
> "-" denotes that the method is not developed for the resolution.
> We also extend NeRV-Diffusion to $256^2$ resolutions with a sublinear increase in model and latent size.
> Please refer to **W2** for full configuration details.
> Overall, our models display superior decoding efficiency over both research and large-scale models.
>
> | Decoders     | #Params | #Params | Latency$\downarrow$ | Latency$\downarrow$ | VRAM$\downarrow$ | VRAM$\downarrow$ |
> |-------------|--------:|--------:|--------------------:|--------------------:|-----------------:|-----------------:|
> |             | $128^2$ | $256^2$ | $128^2$             | $256^2$             | $128^2$          | $256^2$          |
> | Latte       | -       | $49$M   | -                   | $0.288$s            | -                | $5.2$G           |
> | CMD [1]         | $24$M   | $24$M   | $0.129$s            | $1.030$s            | $1.2$G           | $4.1$G           |
> | Open-MAGVIT2 [2] | $226$M  | -       | $0.127$s            | -                   | $45$G            | -                |
> | SD-VAE      | $49$M   | $49$M   | $0.048$s            | $0.260$s            | $1.2$G           | $4.3$G           |
> | SVD-VAE     | $98$M   | $98$M   | $0.094$s            | $0.411$s            | $1.3$G           | $4.5$G           |
> | NeRV-VAE-L  | $55$M   | $64.5$M | $0.032$s            | $0.133$s            | $0.99$G          | $2.6$G           |
>
> **W2: Limited video resolution and length.**
>
> We scale up NeRV-Diffusion to $256^2$ resolution and $128$ frames on UCF.
> Conventional VAEs encode videos into frame-wise feature maps with a fixed spatial downsampling factor, and thus their latent size increases quadratically w.r.t. to RGB resolutions.
> In contrast, NeRV-Diffusion uses INRs as instance-specific decoders, and we append one additional layer/block to the end to perform an extra upsampling to double the output resolution.
> Therefore, our model parameters increase sublinearly, and the neural weight latent size also only needs to increase accordingly.
> Specifically, our NeRV decoder size increases by $17$% as shown in the decoder table W1, and we increase the neural latent token numbe by $25$% for simple alignment of channel-wise parameterization.
> Besides, our GNeRV incorporates skip connections (Section 3.2) so multi-resolution output and joint training are also feasible.
>
> For video length, conventional VAEs increase their frame-wise latent size linearly with a fixed temporal downsampling factor, while our NeRV features smooth native time interpolation and can be trained with large frame intervals as shown in Section 4.5.1.
> In our $128$-frame experiment, we train NeRV-Diffusion at $16$ frames with the downsampling interval of $8$ and sample it with $8\times$ frame interpolation.
> This would be more challenging as no ground truth frames between the anchored frames are seen by our models.
>
> We report the generation performance comparison in below tables.
> In both cases our NeRV-Encoder remains the same configuration.
> NeRV-Diffusion outperforms recent SOTAs with high compactness and easy extensibility.
>
> | Method                 | gFVD$^{256}$$\downarrow$ |
> |------------------------|-------------------------:|
> | VIDM                   | $263$                   |
> | Latte                  | $202$                   |
> | OmniTokenizer          | $191$                   |
> | AR-Diffusion [3]          | $186$                   |
> | HPDM-M [4]                | $143$                   |
> | NeRV-Diffusion-L (Ours)| $140$                   |
>
> | Method                 | gFVD$_{128}$$\downarrow$ |
> |------------------------|-------------------------:|
> | Latte                  | $1157$                  |
> | DIGAN                  | $1103$                  |
> | PVDM [5]                  | $505$                   |
> | VIDM                   | $426$                   |
> | CoordTok [6]              | $369$                   |
> | NeRV-Diffusion-L (Ours)| $366$                   |

---

> ### Author Response · Authors · 2025-11-26
>
> **W3: Long-term semantic consistency of time interpolation.**
>
> We quantitatively measure semantic consistency in terms of object identity and action logic preservation.
> Specifically, we leverage the Subject Consistency and Background Consistency metrics in VBench, which extract the subject and background features via DINO/CLIP, and calculate their pairwise similarities across all frames for appearance preservation.
> Inspired by it, we further employ C3D, a action recognizer pretrained on short UCF videos, and feed it with all sub-clips from the long videos to extract action features.
> We calculate pairwise similarities of the action features across all windows to measure the action logic preservation.
> These metrics don't only rely on consecutive distance but average along the whole video, accurately reflecting the drifting issues for long video generation.
>
> | Long Consistency | VBench$\uparrow$ | VBench$\uparrow$ | Action Sim.$\uparrow$ |
> |------------------|-----------------:|-----------------:|----------------------:|
> |                  | SC               | BC               |                        |
> | Ground Truth     | $0.931$          | $0.956$          | $0.912$               |
> | Ours             | $0.919$          | $0.942$          | $0.901$               |
>
> **W4: Comparison with large-scale video tokenizers.**
>
> LTX VAE and WAN VAE are commercial foundation models that are trained on large-scale data with considerable resources and extensive time.
> As a research paper, we are not able to compete against them but mainly to explore a novel architecture of implicit tokenization and generation that outperforms previous SOTAs under similar settings.
> Though, we list the quantitative comparisons as suggested.
> We also include the VAE of Stable Diffusion (SD) and Stable Video Diffusion (SVD), which are also large-scale foundation models at earlier stages.
>
> | Reconstruction (UCF) | PNSR$\uparrow$ | SSIM$\uparrow$ | LPIPS$\downarrow$ | rFVD$\downarrow$ |
> |----------------------|--------------:|---------------:|------------------:|-----------------:|
> | SD-VAE               | $24.79$       | $0.783$        | $0.068$           | $63$             |
> | SVD-VAE              | $27.24$       | $0.856$        | $0.058$           | $39$             |
> | LTX-VAE              | $31.29$       | $0.909$        | $0.061$           | $32$             |
> | WAN-VAE              | $31.14$       | $0.935$        | $0.022$           | $7$              |
> | NeRV-VAE-L           | $26.63$       | $0.879$        | $0.043$           | $22$             |
>
> **Q1: Inference efficiency comparison.**
>
> We compare the inference speed and peak GPU memory of NeRV-Diffusion with other video generators.
> All results are tested on a single NVIDIA A6000 GPU in bfloat16.
> "-" denotes that the method is not developed for the resolution.
> MAGVIT or Open-MAGVIT2 didn't release their code or checkpoints for generation.
> We also extend NeRV-Diffusion to $256^2$ resolutions with a sublinear increase in model and latent size.
> Please refer to **W2** for full configuration details.
>
> | Generators       | #Params | #Tokens | #Tokens | Steps | Latency$\downarrow$ | Latency$\downarrow$ | VRAM$\downarrow$ | VRAM$\downarrow$ |
> |------------------|--------:|--------:|--------:|------:|--------------------:|--------------------:|-----------------:|-----------------:|
> |                  |         | $128^2$ | $256^2$ |       | $128^2$             | $256^2$             | $128^2$          | $256^2$          |
> | Latte            | $674$M  | -       | $512$   | $250$ | -                   | $37$s               | -                | $4.4$G           |
> | LARP-L           | $343$M  | $1024$  | -       | $1024$| $20$s               | -                   | $1.6$G           | -                |
> | OmniTokenizer    | $650$M  | -       | $1280$  | $5120$| -                   | $139$s              | -                | $4.5$G           |
> | NeRV-Diffusion-L | $467$M  | $128$   | $160$   | $250$ | $6.8$s              | $8.2$s              | $1.8$G           | $2.1$G           |
>
> **Q2: Decoding bottleneck by per-frame forward.**
>
> No, NeRV is a convolutional network and decodes in parallel with all frames as the batch size dimension.
> So there will be no bottleneck when decoding long videos.
>
> **Q3: Encoding and decoding efficiency on 720p.**
>
> We create random data and test the encoding/decoding latency on a $1280\times720$ video.
> The encoding takes approximately $1.9$s, and the decoding takes approximately $1.8$s, both on a single NVIDIA A6000 GPU.
>
> **Q4: Training efficiency comparison.**
>
> We train each of our first-stage NeRV-VAE and second-stage NeRV-Diffusion for $1$ week on $8$ NVIDIA A6000 GPUs.
> We train NeRV-VAE at a small batch size of $32$ for $2$M iterations.
> In comparison, LARP-L-Long is trained for $500$K iterations at batch size of $128$.
> MAGVIT-v2 is trained for $300$K iterations at batch size $256$.
> OmniTokenizer is trained for $1$M iterations on $8$ NVIDIA A100 GPUs for $2$ weeks.

---

> ### Author Response · Authors · 2025-11-26
>
> **Q5: Explanation of multi-head affine modulation.**
>
> In standard configurations of previous INR autoendoers like TransINR and FastNeRV, the bottleneck is formed with two FC layers after the encoder and before the decoder.
> The two FC layers downsample and upsample the token dimensions to form the information bottleneck.
> Our Multi-head affine module contains multiple second FC layers after the bottleneck latent, upsampling it from one to multiple sets of latent tokens.
> In this way, our model is able to modulate all layers in the NeRV decoder with different values that sourced from one bottleneck latent, largely expand its capacity.
>
> **Q6: Token dimension options.**
>
> During our experiments we have ablated different token dimensions, and we noticed a slight performance drop when reducing it (but not as significant as token numbers), so we adopt $128$ while not posting this ablation in the original paper.
> We now list the ablation with NeRV-Diffusion-S configuration here.
>
> | Token Shape          | $128\times32$ | $128\times64$ | $128\times96$ | $128\times128$ |
> |----------------------|-------------:|-------------:|-------------:|--------------:|
> | rFVD$\downarrow$     | $114$        | $99$         | $92$         | $85$          |
> | gFVD$\downarrow$     | $230$        | $212$        | $197$        | $184$         |
>
> We attribute this to that our neural weight latents are holistic representations unlike conventional frame-wise feature maps, and the token dimension doesn't correspond to the RGB color channel.
> Therefore we determine the optimal token dimension via ablation studies.
>
> **Q7: Efficient reconstruction with few tokens.**
>
> Our implicit video tokenizer features its INR decoder dedicated for each data point, i.e. the neural tokens directly serve as decoder parameters instead of input, showing its effectiveness over traditional feature map latents.
> Our multi-head affine module reuses the neural tokens for all layers in the INR decoder, which also enlarges the expressiveness.
>
> **Q8: Additional reconstruction metrics.**
>
> Please refer to **W4** for comparison among large-scale foundation models.
> We also test these metrics for previous work that have released their checkpoints, and report the comparisons below.
>
> | Reconstruction (UCF) | PNSR$\uparrow$ | SSIM$\uparrow$ | LPIPS$\downarrow$ | rFVD$\downarrow$ |
> |----------------------|--------------:|---------------:|------------------:|-----------------:|
> | TATS                 | $22.43$       | $0.765$        | $0.108$           | $162$            |
> | Open-MAGVIT2 [2]     | $25.84$       | $0.862$        | $0.045$           | $16$             |
> | LARP-L               | $27.87$       | $0.891$        | $0.038$           | $20$             |
> | NeRV-VAE-L           | $26.63$       | $0.879$        | $0.043$           | $22$             |
>
> | Reconstruction (K600) | PNSR$\uparrow$ | SSIM$\uparrow$ | LPIPS$\downarrow$ | rFVD$\downarrow$ |
> |-----------------------|--------------:|---------------:|------------------:|-----------------:|
> | LARP-L                | $28.22$       | $0.867$        | $0.035$           | $11$             |
> | NeRV-VAE-L            | $26.45$       | $0.823$        | $0.044$           | $19$             |
>
> It worth noting that the ultimate purpose of NeRV-VAE (and the comparing tokenizers) is to encode RGB videos into smooth latents for high-quality video generation, and we optimize it for best generative quality instead of solely reconstruction faithfulness.
>
> **Q9: Comparison with large-scale models.**
>
> Please refer to **W4**.
>
> **Q10: Token number options.**
>
> For generation, there is usually a balance that too few tokens will lack expressiveness, and too many tokens will harm the smoothness of the latent space, as given the certain amount of each token deviating from standard Gaussian distribution, more tokens will introduce higher total variance and the diffusion model will be harder to converge.
> This applies to all two-stage generation frameworks, given fixed VAE and denoising model sizes.
> We ablated different token numbers in main Table 3(f), and adopted for the optimal generative quality.
> Due to time and resource limit, we didn't ablate on more tokens beyond.

---

> ### Author Response · Authors · 2025-11-26
>
> **Q11: Performance ceiling.**
>
> So far we haven't noticed such a ceiling as NeRV-Diffusion surpasses previous SOTA methods of both diffusion and autoregressive models.
> We attribute this to that our implicit neural weight latent is a holistic representation and leverages the redundancy in videos.
> Meanwhile, the outstanding efficiency of NeRV-Diffusion also indicates its extra potential when scaling up.
> When the RGB resolutions double, NeRV decoder only needs to append one more layer/block to the end for an additional upsampling, and thus the total model size increases sublinearly w.r.t. data resolution.
> Therefore our neural weight latent also increases sublinearly corresponding to the curve of total model size.
> If performance is prioritized over efficiency, NeRV-Diffusion can also further scale up to comparable efficiency for extra performance boost.
>
> **References**:
>
> [1] Efficient Video Diffusion Models via Content-Frame Motion-Latent Decomposition. ICLR 2024.
>
> [2] Open-MAGVIT2: An Open-Source Project Toward Democratizing Auto-regressive Visual Generation. ArXiv 24.09.
>
> [3] AR-Diffusion: Asynchronous Video Generation with Auto-Regressive Diffusion. CVPR 2025.
>
> [4] Hierarchical Patch Diffusion Models for High-Resolution Video Generation. CVPR 2024.
>
> [5] Video Probabilistic Diffusion Models in Projected Latent Space. CVPR 2023.
>
> [6] Efficient Long Video Tokenization via Coordinate-based Patch Reconstruction. ArXiv 24.11.
>
> If you find our response helpful, please kindly consider raising your ratings.
> If you have more concerns, please feel free to let us know and we will promptly update.
> We sincerely appreciate your valuable discussion and feedback!

---

### Official Review · Reviewer_WJ85 · 2025-10-28

**Soundness:** 2
**Presentation:** 3
**Contribution:** 2
**Rating:** 4
**Confidence:** 5

**Summary:**

The paper proposes a two-stage, INR-based generative pipeline: (1) a hypernetwork tokenizer encodes a video into a compact latent vector whose entries are used directly as the convolution kernels of a small NeRV decoder; (2) a Diffusion Transformer (DiT) denoises in this weight-token space to generate new videos. The authors have taken measures including KL bottleneck after the encoder, Gaussian bottleneck, channel-wise parameterization to ensure the weights follow (multivariate) normal distribution and facilitate smooth interpolations. On UCF-101 class-conditioned generation and Kinetics-600 frame prediction, the method reports competitive (g)FVD at 128×128, 16 frames, with fast per-frame decoding thanks to the tiny instance-specific decoder. It also shows smooth time and weight-space interpolations.

**Strengths:**

- The idea of generating leveraging the lightweightness and interpolation ability of INRs to produce videos with good spatiotemporal coherence (without having to rely on cross-attention) is interesting.
- The model is compact and associated with fast decoding. Requiring only 128 tokens, a very small decoder, and a reasonable-sized DiT generator, the model still delivers pretty impressive performance.
- The engineering efforts that make the weight-space diffusion work all make intuitive sense and well supported by the ablation results.

**Weaknesses:**

- **Narrow scale & task scope.** Though the use of UCF-101 and K600 is standard, all experiments are conducted only at 128x128 resolution with 16 frames, which is acceptable for early proofs-of-concept but could be disadvantageous for those larger baselines, as this setup is too simple to expose scalability at higher resolutions & longer videos.
- **Heavy training.** The two-stage system requires longer training times and increases engineering burden, and it is expected that scaling up this framework might be even heavier; further, the DiT is still sizeable so it is hard to determine the actual runtime speed up saved.
- **Gaussianity is not verified.** The paper doesn't show diagnostics (e.g., per-layer marginal tests, normality plots, etc.) to verify how Gaussian the INR weights actually are, and under what conditions would the Gaussianity assumption be counter-effective?
- **Limited metrics.** The authors should consider reporting more metrics (e.g., tLPIPS, or some other more well-rounded metrics, like those from VBench or WorldScore), especially consider the performance is not reported for the K600 dataset.

I think the paper has good novelty and the combination of diffusion (or generative models in general) with INRs for video generation/synthesis could be of great interests to the community. However, I am slightly concerned with how extendable & scalable this method is and how well it performs after more comprehensive evaluations, but I am willing to raise my score if these concerns are addressed by the authors' response.

**Questions:**

- What might be the failure modes for the Gaussianity-diffusion trade-off case? Would this be harder to guarantee for larger decoder/videos or videos with more complicated spatiotemporal dynamics?
- To scale up to higher resolutions, which one of NeRV capacity, weight latent size, DiT training, would be the first to break? Did you try increasing token size or decoder width beyond the ablation settings?

---

> ### Author Response · Authors · 2025-11-26
>
> We appreciate your acknowledgment of our interesting idea and the lightweightness and interpolation ability of our INR generation, as well as its efficiency and performance.
> We address your concerns and questions as below:
>
> **W1: Video resolution and length scales.**
>
> We scale up NeRV-Diffusion to $256^2$ resolution and $128$ frames on UCF.
> Conventional VAEs encode videos into frame-wise feature maps with a fixed spatial downsampling factor, and thus their latent size increases quadratically w.r.t. to RGB resolutions.
> In contrast, NeRV-Diffusion uses INRs as instance-specific decoders, and we append one additional layer/block to the end to perform an extra upsampling to double the output resolution.
> Therefore, our model parameters increase sublinearly, and the neural weight latent size also only needs to increase accordingly.
> Specifically, our NeRV decoder size increases by $17$% as shown in the decoder table W1, and we increase the neural latent token numbe by $25$% for simple alignment of channel-wise parameterization.
> Besides, our GNeRV incorporates skip connections (Section 3.2) so multi-resolution output and joint training are also feasible.
>
> For video length, conventional VAEs increase their frame-wise latent size linearly with a fixed temporal downsampling factor, while our NeRV features smooth native time interpolation and can be trained with large frame intervals as shown in Section 4.5.1.
> In our $128$-frame experiment, we train NeRV-Diffusion at $16$ frames with the downsampling interval of $8$ and sample it with $8\times$ frame interpolation.
> This would be more challenging as no ground truth frames between the anchored frames are seen by our models.
>
> We report the generation performance comparison in below tables.
> In both cases our NeRV-Encoder remains the same configuration.
> NeRV-Diffusion outperforms recent SOTAs with high compactness and easy extensibility.
>
> | Method                 | gFVD$^{256}$$\downarrow$ |
> |------------------------|-------------------------:|
> | VIDM                   | $263$                   |
> | Latte                  | $202$                   |
> | OmniTokenizer          | $191$                   |
> | AR-Diffusion [3]          | $186$                   |
> | HPDM-M [4]                | $143$                   |
> | NeRV-Diffusion-L (Ours)| $140$                   |
>
> | Method                 | gFVD$_{128}$$\downarrow$ |
> |------------------------|-------------------------:|
> | Latte                  | $1157$                  |
> | DIGAN                  | $1103$                  |
> | PVDM [5]                  | $505$                   |
> | VIDM                   | $426$                   |
> | CoordTok [6]              | $369$                   |
> | NeRV-Diffusion-L (Ours)| $366$                   |
>
> **W2: Two-stage generative framework.**
>
> A two-stage framework has been widely adopted for almost all large-scale generative models.
> Latent Diffusion Models (LDMs, or Stable Diffusions, SDs) encodes RGB data into continuous latents and train diffusion models on the latents.
> Vector-Quantized GANs (VQGANs) also encodes RGB data into discrete codes and train autoregressive (AR) models on the codes.
> Compared to single-stage generation models such as GANs, two-stage frameworks are known for their stability and efficiency especially when scaled up to large-scale models and data.
>
> We adopt a $457$M DiT model following the previous SOTA methods with similar generator parameters for fair comparison.
> We compare the inference speed and peak GPU memory of both stages.
> All results are tested on a single NVIDIA A6000 GPU in bfloat16.
> "-" denotes that the method is not developed for the resolution.
> Our models have far less latency and VRAM footprint compared to recent methods, demonstrating the efficiency of our implicit framework.
>
> | Decoders     | #Params | #Params | Latency$\downarrow$ | Latency$\downarrow$ | VRAM$\downarrow$ | VRAM$\downarrow$ |
> |-------------|--------:|--------:|--------------------:|--------------------:|-----------------:|-----------------:|
> |             | $128^2$ | $256^2$ | $128^2$             | $256^2$             | $128^2$          | $256^2$          |
> | Latte       | -       | $49$M   | -                   | $0.288$s            | -                | $5.2$G           |
> | CMD [5] | $24$M   | $24$M   | $0.129$s            | $1.030$s            | $1.2$G           | $4.1$G           |
> | Open-MAGVIT2 [6] | $226$M  | -       | $0.127$s            | -                   | $45$G            | -                |
> | NeRV-VAE-L  | $55$M   | $64.5$M | $0.032$s            | $0.133$s            | $0.99$G          | $2.6$G           |
>
> **(Generator comparison continues in the next comment.)**

---

> ### Author Response · Authors · 2025-11-26
>
> **(Continued generator comparison for W2.)**
>
> | Generators       | #Params | #Tokens | #Tokens | Steps | Latency$\downarrow$ | Latency$\downarrow$ | VRAM$\downarrow$ | VRAM$\downarrow$ |
> |------------------|--------:|--------:|--------:|------:|--------------------:|--------------------:|-----------------:|-----------------:|
> |                  |         | $128^2$ | $256^2$ |       | $128^2$             | $256^2$             | $128^2$          | $256^2$          |
> | Latte            | $674$M  | -       | $512$   | $250$ | -                   | $37$s               | -                | $4.4$G           |
> | LARP-L           | $343$M  | $1024$  | -       | $1024$| $20$s               | -                   | $1.6$G           | -                |
> | OmniTokenizer    | $650$M  | -       | $1280$  | $5120$| -                   | $139$s              | -                | $4.5$G           |
> | NeRV-Diffusion-L | $467$M  | $128$   | $160$   | $250$ | $6.8$s              | $8.2$s              | $1.8$G           | $2.1$G           |
>
> **W3: Gaussianity diagnostics.**
>
> We conduct marginal tests and normality plots on our encoded neural weight latent as suggested, and compare it to LDM/SD's VAE latent.
> The visualizations are presented in our revised paper Appendix Figure. A8.
>
> | Gaussianity | mean        | std       | skew       | kurtosis   |
> |-------------|------------:|----------:|-----------:|-----------:|
> | LDM/SD-VAE  | $-0.0165$   | $5.2505$  | $-0.4756$  | $0.1693$   |
> | NeRV-VAE-L  | $-0.0036$   | $0.9979$  | $-0.0001$  | $0.2568$   |
>
> Gaussianity makes smooth latent spaces where diffusion models are easier to be trained to map Gaussian noise to target latent, while it is the common trade-off for all VAEs that an over high Gaussianity harms the expressiveness and fidelity.
> We ablate different KL loss scales on NeRV-Diffusion-S configuration in the table below.
>
> | KL Loss Scale          | $1\times10^{-6}$ | $5\times10^{-6}$ | $1\times10^{-5}$ | $5\times10^{-5}$ |
> |------------------------|-------------:|-------------:|-------------:|-------------:|
> | rFVD$\downarrow$       | $73$         | $82$         | $85$         | $107$        |
> | gFVD$\downarrow$       | $198$        | $186$        | $184$        | $202$        |

---

> ### Author Response · Authors · 2025-11-26
>
> **W4: Limited metrics on UCF and K600.**
>
> The use of FVD for quantifying both reconstruction and generation quality on UCF and K600 datasets is a standard pipeline that have been widely practiced by SOTA work in recent top venues including MAGVITs, OmniTokenizer, HPDM, LARP and AR-Diffusion etc.
> They didn't report additional metrics and we initially followed their protocols for fair comparison.
>
> We reported rFVD and gFVD on K600 in main Table 1.
> Now we extend our reconstruction metrics to PNSR, SSIM and LPIPS between the output and input videos, and extend our generation metrics to cross-frame LPIPS (fLPIPS) and C3D/I3D-based Inception Score (IS) on the output videos.
> As suggested, we also evaluate the non-text subsets of VBench on generators, including Subject Consistency (SC), Background Consistency (BC), Temporal Flickering (TF), Motion Smoothness (MS), Dynamic Degree (DD), Aesthetic Quality (AQ) and Imaging Quality (IQ).
> For comparing methods, we use their pretrained checkpoints for inference.
>
> It worth noting that the ultimate purpose of NeRV-VAE (and the comparing tokenizers) is to encode RGB videos into smooth latents for high-quality video generation, and we optimize it for best generative quality instead of solely reconstruction faithfulness.
> Our models achieve SOTA generation performance on UCF across all dimensions, and reach comparable SOTA performance on K600 frame prediction.
>
> | Reconstruction (UCF) | PNSR$\uparrow$ | SSIM$\uparrow$ | LPIPS$\downarrow$ | rFVD$\downarrow$ |
> |----------------------|--------------:|---------------:|------------------:|-----------------:|
> | TATS                 | $22.43$       | $0.765$        | $0.108$           | $162$            |
> | Open-MAGVIT2 [2]     | $25.84$       | $0.862$        | $0.045$           | $16$             |
> | LARP-L               | $27.87$       | $0.891$        | $0.038$           | $20$             |
> | NeRV-VAE-L           | $26.63$       | $0.879$        | $0.043$           | $22$             |
>
> | Generation (UCF)   | fLPIPS$\updownarrow$ | IS$\uparrow$ | VBench$\uparrow$ | VBench$\uparrow$ | VBench$\uparrow$ | VBench$\uparrow$ | VBench$\uparrow$ | VBench$\uparrow$ | VBench$\uparrow$ | gFVD$\downarrow$ |
> |--------------------|---------------------:|------------:|-----------------:|-----------------:|-----------------:|-----------------:|-----------------:|-----------------:|-----------------:|-----------------:|
> |                    |                      |             | SC               | BC               | TF               | MS               | DD               | AQ               | IQ               |                  |
> | Ground Truth       | $0.031$             | $86.24$     | $0.954$          | $0.977$          | $0.981$          | $0.988$          | $0.282$          | $0.410$          | $0.443$          | $0$              |
> | TATS               | $0.048$             | $68.79$     | $0.910$          | $0.958$          | $0.978$          | $0.980$          | $0.314$          | $0.348$          | $0.437$          | $332$            |
> | VIDM [4]           | -                   | $64.17$     | -                | -                | -                | -                | -                | -                | -                | $263$            |
> | VideoDiffusion     | -                   | $80.03$     | -                | -                | -                | -                | -                | -                | -                | $173$            |
> | LARP-L             | $0.025$             | $68.79$     | $0.955$          | $0.977$          | $0.985$          | $0.991$          | $0.218$          | $0.398$          | $0.393$          | $102$            |
> | NeRV-Diffusion-L   | $0.028$             | $82.17$     | $0.958$          | $0.978$          | $0.983$          | $0.989$          | $0.262$          | $0.392$          | $0.433$          | $97$             |
>
> **(K600 results continue in the next comment.)**

---

> ### Author Response · Authors · 2025-11-26
>
> **(Continued K600 results for W4.)**
>
> | Reconstruction (K600) | PNSR$\uparrow$ | SSIM$\uparrow$ | LPIPS$\downarrow$ | rFVD$\downarrow$ |
> |-----------------------|--------------:|---------------:|------------------:|-----------------:|
> | LARP-L                | $28.22$       | $0.867$        | $0.035$           | $11$             |
> | NeRV-VAE-L            | $26.45$       | $0.823$        | $0.044$           | $19$             |
>
> | Generation (K600)   | fLPIPS$\updownarrow$ | IS$\uparrow$ | VBench$\uparrow$ | VBench$\uparrow$ | VBench$\uparrow$ | VBench$\uparrow$ | VBench$\uparrow$ | VBench$\uparrow$ | VBench$\uparrow$ | gFVD$\downarrow$ |
> |---------------------|---------------------:|------------:|-----------------:|-----------------:|-----------------:|-----------------:|-----------------:|-----------------:|-----------------:|-----------------:|
> |                     |                      |             | SC               | BC               | TF               | MS               | DD               | AQ               | IQ               |                  |
> | Ground Truth        | $0.031$             | $31.20$     | $0.950$          | $0.945$          | $0.978$          | $0.988$          | $0.266$          | $0.406$          | $0.461$          | $0$              |
> | LARP-L              | $0.029$             | $23.51$     | $0.933$          | $0.973$          | $0.981$          | $0.989$          | $0.292$          | $0.337$          | $0.293$          | $17$             |
> | NeRV-Diffusion-L    | $0.034$             | $27.09$     | $0.928$          | $0.967$          | $0.977$          | $0.979$          | $0.354$          | $0.392$          | $0.432$          | $22$             |
>
> **Q1: Gaussianity trade-off.**
>
> An over low Gaussianity will boost reconstruction but harm generation, while an over high Gaussianity will reduce the gap between reconstruction and generation but harm their absolute values.
> Please refer to **W3** for more details.
>
> All latent diffusion models need to balance between Gaussianity and expressiveness of the latent space, and almost all large-scale foundation diffusion models, such as T2I, T2V, etc. follow this framework.
> The expressiveness and dynamics of latent diffusion models for larger-scale and more complex data will not be limited by the KL trade-off.
> Without VAE, pixel diffusion models will have to process raw RGB data and is harder to scale up.
> Without the KL constraint, diffusion models will be hard to converge on the unbounded latent values.
>
> **Q2: Further scaling up.**
>
> When the RGB resolutions double, NeRV decoder only needs to append one more layer/block to the end for an additional upsampling, and thus the total model size increases sublinearly w.r.t. data resolution.
> Therefore our neural weight latent also increases sublinearly corresponding to the curve of total model size.
> Our explorations started from compact NeRV decoders and we found that scaling up NeRV decoders would considerably improve the performance.
> We have ablated neural weight latent size in main Table 2(f), and it shows that a too big latent size will be harder for DiT of the same size to train on.
> However this is the common pattern for all VAEs and latent diffusions, as the upper bound of latent size would be the raw RGB size, and LDMs usually outperform pixel diffusions.
> Due to resource limit, we haven't been able to further scale up the latent or decoder size as our current experiments have shown scaling-up boost.
>
> **References**:
>
> [1] AR-Diffusion: Asynchronous Video Generation with Auto-Regressive Diffusion. CVPR 2025.
>
> [2] Hierarchical Patch Diffusion Models for High-Resolution Video Generation. CVPR 2024.
>
> [3] Video Probabilistic Diffusion Models in Projected Latent Space. CVPR 2023.
>
> [4] Efficient Long Video Tokenization via Coordinate-based Patch Reconstruction. ArXiv 24.11.
>
> [5] Efficient Video Diffusion Models via Content-Frame Motion-Latent Decomposition. ICLR 2024.
>
> [6] Open-MAGVIT2: An Open-Source Project Toward Democratizing Auto-regressive Visual Generation. ArXiv 24.09.

---

> > ### Author Response · Authors · 2025-11-26
> >
> > If you find our response helpful, please kindly consider raising your ratings.
> > If you have more concerns, please feel free to let us know and we will promptly update.
> > We sincerely appreciate your valuable discussion and feedback!

---

### Official Review · Reviewer_FaBD · 2025-11-01

**Soundness:** 3
**Presentation:** 3
**Contribution:** 3
**Rating:** 6
**Confidence:** 3

**Summary:**

NeRV-Diffusion is a novel two-stage generative framework for video synthesis that operates by generating the parameters of an Implicit Neural Representation (INR). Instead of synthesizing videos in pixel space or a traditional latent feature space, it first uses a hypernetwork-based autoencoder to "tokenize" videos, compressing them into a compact set of latent INR weights. This latent space is explicitly regularized to follow a Gaussian distribution. In the second stage, an Implicit Diffusion Transformer (DiT) is trained to model this latent space. To generate a new video, the DiT denoises a random noise vector into a new set of valid INR weights. These weights are then used to parameterize a convolutional INR (a "Generative NeRV"), which can decode the full-length video by taking continuous frame indices as input. The model achieves performance comparable to state-of-the-art non-implicit models on benchmarks like UCF-101 and Kinetics-600, while offering significant efficiency advantages and smooth interpolation properties.

**Strengths:**

1. By representing the video holistically as a single neural network, the model obviates the need for computationally expensive cross-frame attentions in the denoising network. This directly addresses a major computational bottleneck present in traditional frame-wise video diffusion models.
2. The framework generates a compact, instance-specific decoder (the INR) rather than relying on a large, shared, universal decoder (like a VAE). INRs are known for being fast at the decoding (rendering) stage, making the visualization of generated videos computationally efficient.
3. The model can be trained on sparse frames but generate smooth, high-frame-rate videos simply by sampling more frame indices at inference time (e.g., generating 128 frames from a 16-frame training setup).
4. Traditional tokenizers that create frame-wise feature maps lead to redundant representations by ignoring inter-frame coherence. NeRV-Diffusion's approach of encoding the entire video into a single set of parameters is a much more compact and holistic representation.
5. Despite being a new paradigm, the model demonstrates high-quality results. It significantly outperforms previous INR-based generative models (like DIGAN) and achieves comparable performance to non-implicit SOTA models (like MAGVITV2-AR) on standard gFVD benchmarks.

**Weaknesses:**

1. The framework is not end-to-end. It relies on a two-stage pipeline: (1) training the NeRV autoencoder, and (2) training the diffusion model on the frozen latent space. The final generation quality is fundamentally bottlenecked by the fidelity and "Gaussian-ness" of the latent space produced by the first stage.
2. The latent space consists of neural network weights, which are highly abstract and have no intuitive spatiotemporal or semantic meaning. This makes granular, controllable editing extremely difficult, if not impossible, without retraining.
3. The core challenge is forcing the INR weight space to be both highly expressive (to reconstruct complex videos) and simple enough to be modeled by a Gaussian distribution (for diffusion). The paper's numerous architectural modifications (multi-head affine, channel-wise parameterization et. al.) highlight the difficulty of balancing this trade-off.
4. The experiments are conducted on low-resolution (128) and short (16-frame) videos. It is unclear how this framework scales to high-definition (HD/4K) or long-duration videos. A higher-resolution INR would require vastly more parameters, creating a much larger token sequence for the DiT to generate, which is a significant challenge.

**Questions:**

1. How does the performance (training stability, gFVD) and computational cost scale with respect to (a) video resolution (e.g. 512) and (b) video duration (e.g., 256 frames)? What is the growth rate of the INR parameter count (the token sequence length for the DiT)?
2. Given the abstract nature of the weight space, what are the prospects for conditional generation beyond simple class labels or frame-gaps? Could this framework be guided by text, depth maps, or semantic layouts?
3. How sensitive is the final generation quality to the fidelity of the first-stage autoencoder? How important is the D_KL loss weight in balancing reconstruction fidelity vs. a smooth, generative latent space?

---

> ### Author Response · Authors · 2025-11-26
>
> We appreciate your acknowledgment of the computational efficiency of our holistic neural representations and instance-specific INR decoders, as well as the flexibility of its time interpolation ability and the overall performance and quality.
> We address your concerns and questions as below:
>
> **W1: Two-stage framework and Gaussianity**
>
> A two-stage framework has been widely adopted for almost all large-scale generative models.
> Latent Diffusion Models (LDMs, or Stable Diffusions, SDs) encodes RGB data into continuous latents and train diffusion models on the latents.
> Vector-Quantized GANs (VQGANs) also encodes RGB data into discrete codes and train autoregressive (AR) models on the codes.
> They both need to balance between the latent space smoothness (KL distance in VAEs and codebook entropy in VQGANs) and reconstruction quality when training the first-stage tokenizers, which determine the upper bound of the generation quality.
> Compared to single-stage generation models such as GANs, two-stage frameworks are known for their stability and efficiency especially when scaled up to large-scale models and data.
>
> **W2: Granularity and controllability of neural network weights**
>
> Neural weight latents form up the parameters of the INR decoder, taking in unified spatiotemporal input embeddings to render pixel frames.
> Although it is holistic and different from conventional frame-wise feature maps, on the other hand it still performs convolutions, i.e. matrix multiplications over the spatiotemporal input grid (in the opposite positions), and thus still maintains spatiotemporal information.
> This is a major difference of our implicit representations between discrete holistic video latents such as LARP.
> We demonstrate the smoothness of our neural network weight space in Section 4.5.2 (Appendix Section C) via direct interpolations between them.
>
> We further design another probing experiment to validate the spatiotemporal association between our neural weight latents and pixel frames.
> We crop RGB videos into random clips with random spatial and temporal ranges, and encode them using various video encoders.
> For each video encoder, we perform K-Means on its encoded latents, with the raw video sources as the class labels.
> We calculate the purity and normalized mutual information (NMI) between the latent clusters and their real labels, i.e. which raw video the corresponding clip is cropped from.
> The experiment shows that our neural weight latents have spatiotemporal relations with pixel operations, indicating its potential to facilitate granular editing.
>
> | Granularity | Purity$\uparrow$ | NMI$\uparrow$ |
> |------------|-----------------:|--------------:|
> | LDM/SD-VAE | $0.484$          | $0.753$       |
> | LARP       | $0.285$          | $0.591$       |
> | Ours       | $0.366$          | $0.656$       |
>
> **W3: Trade-off between expressiveness and Gaussianity**
>
> As we illustrate in W1, all latent diffusion models demand a VAE to encode RGB data into smooth latent, and all VAEs need to balance between the KL and reconstruction loss in training.
> An autoencoder without variational (KL) constraint cannot be used to train diffusions.
> In our work, we are able to manage this balance well and provide a stable and clear recipe with high quality.
> Our proposed modules, such as multi-head affine and channel-wise parameterization, are mainly designed for adapting and enhancing the NeRV interface.
> These components exist in prior work such as StyleGAN, DIGAN, TransINR and FastNeRV, and we make simple yet effective upgrades to largely lift their expressiveness.

---

> ### Author Response · Authors · 2025-11-26
>
> **W4: Data scale and parameters.**
>
> We scale up NeRV-Diffusion to $256^2$ resolution and $128$ frames on UCF.
> Conventional VAEs encode videos into frame-wise feature maps with a fixed spatial downsampling factor, and thus their latent size increases quadratically w.r.t. to RGB resolutions.
> In contrast, NeRV-Diffusion uses INRs as instance-specific decoders, and we append one additional layer/block to the end to perform an extra upsampling to double the output resolution.
> Therefore, our model parameters increase sublinearly, and the neural weight latent size also only needs to increase accordingly.
> Specifically, our NeRV decoder size increases by $17$% as shown in the decoder table W1, and we increase the neural latent token numbe by $25$% for simple alignment of channel-wise parameterization.
> Besides, our GNeRV incorporates skip connections (Section 3.2) so multi-resolution output and joint training are also feasible.
>
> For video length, conventional VAEs increase their frame-wise latent size linearly with a fixed temporal downsampling factor, while our NeRV features smooth native time interpolation and can be trained with large frame intervals as shown in Section 4.5.1.
> In our $128$-frame experiment, we train NeRV-Diffusion at $16$ frames with the downsampling interval of $8$ and sample it with $8\times$ frame interpolation.
> This would be more challenging as no ground truth frames between the anchored frames are seen by our models.
>
> We report the generation performance comparison in below tables.
> In both cases our NeRV-Encoder remains the same configuration.
> NeRV-Diffusion outperforms recent SOTAs with high compactness and easy extensibility.
>
> | Method                 | gFVD$^{256}$$\downarrow$ |
> |------------------------|-------------------------:|
> | VIDM                   | $263$                   |
> | Latte                  | $202$                   |
> | OmniTokenizer          | $191$                   |
> | AR-Diffusion [1]          | $186$                   |
> | HPDM-M [2]                | $143$                   |
> | NeRV-Diffusion-L (Ours)| $140$                   |
>
> | Method                 | gFVD$_{128}$$\downarrow$ |
> |------------------------|-------------------------:|
> | Latte                  | $1157$                  |
> | DIGAN                  | $1103$                  |
> | PVDM [3]                  | $505$                   |
> | VIDM                   | $426$                   |
> | CoordTok [4]              | $369$                   |
> | NeRV-Diffusion-L (Ours)| $366$                   |
>
> **Q1: Scaling up data and models.**
>
> We conduct scaling-up experiments on $256^2$ resolutions and $128$ frames in **W4**.
> NeRV-Diffusion presents leading performance with sublinear parameter and latent space increase.
> A two-stage generation framework is known to have consistent stability for larger-scale models and data, and we also observe our scaling-up experiments steady.
> For more details please refer to **W4**.
>
> We compare the inference speed and peak GPU memory at different resolutions.
> All results are tested on a single NVIDIA A6000 GPU in bfloat16.
> "-" denotes that the method is not developed for the resolution.
> Our models have far less latency and VRAM footprint with sublinear increase compared to recent methods, demonstrating the efficiency of our implicit framework.
>
> | Decoders     | #Params | #Params | Latency$\downarrow$ | Latency$\downarrow$ | VRAM$\downarrow$ | VRAM$\downarrow$ |
> |-------------|--------:|--------:|--------------------:|--------------------:|-----------------:|-----------------:|
> |             | $128^2$ | $256^2$ | $128^2$             | $256^2$             | $128^2$          | $256^2$          |
> | Latte       | -       | $49$M   | -                   | $0.288$s            | -                | $5.2$G           |
> | CMD [5] | $24$M   | $24$M   | $0.129$s            | $1.030$s            | $1.2$G           | $4.1$G           |
> | Open-MAGVIT2 [6] | $226$M  | -       | $0.127$s            | -                   | $45$G            | -                |
> | NeRV-VAE-L  | $55$M   | $64.5$M | $0.032$s            | $0.133$s            | $0.99$G          | $2.6$G           |
>
> **(Generator comparison continue in the next comment.)**

---

> ### Author Response · Authors · 2025-11-26
>
> **(Continued generator comparison for Q1.)**
>
> | Generators       | #Params | #Tokens | #Tokens | Steps | Latency$\downarrow$ | Latency$\downarrow$ | VRAM$\downarrow$ | VRAM$\downarrow$ |
> |------------------|--------:|--------:|--------:|------:|--------------------:|--------------------:|-----------------:|-----------------:|
> |                  |         | $128^2$ | $256^2$ |       | $128^2$             | $256^2$             | $128^2$          | $256^2$          |
> | Latte            | $674$M  | -       | $512$   | $250$ | -                   | $37$s               | -                | $4.4$G           |
> | LARP-L           | $343$M  | $1024$  | -       | $1024$| $20$s               | -                   | $1.6$G           | -                |
> | OmniTokenizer    | $650$M  | -       | $1280$  | $5120$| -                   | $139$s              | -                | $4.5$G           |
> | NeRV-Diffusion-L | $467$M  | $128$   | $160$   | $250$ | $6.8$s              | $8.2$s              | $1.8$G           | $2.1$G           |
>
> **Q2: General condition controls.**
>
> Our implicit diffusion model follows standard diffusion framework but only switches to neural weight latents, and thus can be trained with various condition types.
> Additional condition input can be integrated with standard cross attention so that different modalities other than similar videos or images are supported.
> We follow the prior work that developed novel tokenizers to majorly test on the foundation tasks.
> Due to time limit and resource scale, we extend NeRV-Diffusion to edge map conditioned video generation on K600.
> Previous work didn't evaluate this protocol, and we present our qualitative results in the revised paper Appendix Figure A7(f).
> This demonstrates the extensibility of NeRV-Diffusion adapting to fine-grained spatiotemporal condition controls.
>
> **Q3: Bottleneck balance between two stages.**
>
> In all two-stage frameworks, the reconstruction fidelity of the first-stage VAE determines the upper bound of the generation quality of the second-stage generator.
> Given other factors (e.g. KL loss scale) fixed, the generation quality will follow the change of reconstruction fidelity.
> Main Table 1 shows that for different model size configurations (NeRV-Diffusion-S, -B, -L), our gFVD changes following rFVD.
> The gap between gFVD and rFVD is controlled by the KL loss scale.
> A high KL loss scale will harm rFVD but reduce the gap between gFVD and rFVD, while a low KL loss scale will improve rFVD but enlarge the gap between gFVD and rFVD.
> We ablate its impact on NeRV-Diffusion-S configuration in the table below.
> All VAEs need to balance on this factor, and we provide a clear and stable recipe.
>
> | KL Loss Scale     | $1\times10^{-6}$ | $5\times10^{-6}$ | $1\times10^{-5}$ | $1\times10^{-5}$ |
> |-------------------|------------:|-------------:|-------------:|-------------:|
> | rFVD              | $73$        | $82$         | $85$         | $107$        |
> | gFVD              | $198$       | $186$        | $184$        | $202$        |
>
> Moreover, main Table 1 shows that different methods have their own optimal gFVD-rFVD gap, corresponding to their optimal balance between the reconstruction fidelity and latent space smoothness constraint.
> NeRV-Diffusion features a relatively low gFVD-rFVD gap at all scales, indicating the smoothness of our implicit latent space.
> We also attribute this to that conventional frame-wise tokenizers rely on input videos to guide their motion flow for reconstruction, while usually needing temporal attentions to constrain temporal consistancy in the generator, which create extra challenge and thus reconstruction-generation gap.
> NeRV-Diffusion highlights its holistic video representations without cross-frame regularization in either VAE and the implicit DiT, leading to consistent performance across reconstruction and generation.
>
> **References**:
>
> [1] AR-Diffusion: Asynchronous Video Generation with Auto-Regressive Diffusion. CVPR 2025.
>
> [2] Hierarchical Patch Diffusion Models for High-Resolution Video Generation. CVPR 2024.
>
> [3] Video Probabilistic Diffusion Models in Projected Latent Space. CVPR 2023.
>
> [4] Efficient Long Video Tokenization via Coordinate-based Patch Reconstruction. ArXiv 24.11.
>
> [5] Efficient Video Diffusion Models via Content-Frame Motion-Latent Decomposition. ICLR 2024.
>
> [6] Open-MAGVIT2: An Open-Source Project Toward Democratizing Auto-regressive Visual Generation. ArXiv 24.09.

---

> > ### Author Response · Authors · 2025-11-26
> >
> > If you find our response helpful, please kindly consider raising your ratings.
> > If you have more concerns, please feel free to let us know and we will promptly update.
> > We sincerely appreciate your valuable discussion and feedback!

---

### Official Review · Reviewer_kMwA · 2025-11-02

**Soundness:** 2
**Presentation:** 2
**Contribution:** 1
**Rating:** 2
**Confidence:** 3

**Summary:**

This paper proposes NeRV-Diffusion, which synthesizes videos by generating implicit neural representation (INR) weights through diffusion models. The method first uses a hypernetwork-based tokenizer to encode videos into neural network parameters, then uses a diffusion transformer to denoise and generate in the weight space. The core idea is to represent videos holistically as a set of INR weights, rather than frame-by-frame feature maps.

**Strengths:**

The idea of shifting from generating pixels to generating neural network weights is unique in the video generation field. Although there is precedent for INR generation (DIGAN 2022), applying diffusion models to video INR weight space still has exploratory significance.

The combined design of KL bottleneck, multi-head affine modulation, and channel-level parameterization to some extent solves the contradiction between weight distribution regularization and expressive capability.

**Weaknesses:**

Only compared decoder parameter count (3.5M-55M) and token count (128), missing actual inference speed.

Evaluation metrics are too singular, only using FVD as one metric.

Video generation resolution is low and length is short, all experiments only at 128×128, unable to prove the method can scale to higher resolutions.

The dataset scale used is small and old, unable to verify the method's performance in real large-scale scenarios.

Comparison methods and baselines are outdated, missing comparisons with latest methods such as wan2.1, Self-Forcing, etc., and method metrics are too few, moreover not SOTA, advantages are not obvious.

Lacks qualitative comparison results display and analysis with other methods.

Interpolation experiments lack quantitative metric verification, only on class-conditioned and frame prediction two tasks, unable to prove the method's generalizability, such as text-to-video, image-to-video, video editing, unconditional generation.

The core paradigm (generating INR weights) comes from DIGAN, improvements are mainly engineering tricks (KL constraint, multi-head modulation, scheduled sampling, etc.).

**Questions:**

Please address the problems in the weakness section.

---

> ### Author Response · Authors · 2025-11-26
>
> We appreciate your acknowledgment of our "unique" idea and the effectiveness of our proposed approaches.
> We also want to further explain that, the balance between latent distribution regularization and expressive capability is common in all two-stage generation frameworks, including all latent diffusion models (LDMs) and autoregressive (AR) models.
> This trade-off between compactness and richness always exists, and we are able to manage it well and provide a stable and clear recipe with high quality.
> Our proposed modules, such as multi-head affine and channel-wise parameterization, are mainly designed for adapting and enhancing the NeRV interface.
> These components exist in prior work such as StyleGAN and TransINR, and we make effective upgrades to largely lift their performance.
>
> We address your concerns and questions as below:
>
> **W1: Inference speed comparison.**
>
> We compare the inference speed and peak GPU memory of NeRV-Decoder and NeRV-Diffusion with other video decoders and generators.
> All results are tested on a single NVIDIA A6000 GPU in bfloat16.
> "-" denotes that the method is not developed for the resolution.
> We also extend NeRV-Diffusion to $256^2$ resolutions with a sublinear increase in model and latent size.
> Please refer to **W3** for full configuration details.
>
> Our models cost far less latency and VRAM footprint in both decoding and generation stages.
> It demonstrates the superior efficiency of our implicit framework, especially due to obviating temporal attentions by reusing the same set of parameters to decode all frames with redundancy.
>
> | Decoders     | #Params | #Params | Latency$\downarrow$ | Latency$\downarrow$ | VRAM$\downarrow$ | VRAM$\downarrow$ |
> |-------------|--------:|--------:|--------------------:|--------------------:|-----------------:|-----------------:|
> |             | $128^2$ | $256^2$ | $128^2$             | $256^2$             | $128^2$          | $256^2$          |
> | Latte       | -       | $49$M   | -                   | $0.288$s            | -                | $5.2$G           |
> | CMD [1] | $24$M   | $24$M   | $0.129$s            | $1.030$s            | $1.2$G           | $4.1$G           |
> | Open-MAGVIT2 [2] | $226$M  | -       | $0.127$s            | -                   | $45$G            | -                |
> | NeRV-VAE-L  | $55$M   | $64.5$M | $0.032$s            | $0.133$s            | $0.99$G          | $2.6$G           |
>
> | Generators       | #Params | #Tokens | #Tokens | Steps | Latency$\downarrow$ | Latency$\downarrow$ | VRAM$\downarrow$ | VRAM$\downarrow$ |
> |------------------|--------:|--------:|--------:|------:|--------------------:|--------------------:|-----------------:|-----------------:|
> |                  |         | $128^2$ | $256^2$ |       | $128^2$             | $256^2$             | $128^2$          | $256^2$          |
> | Latte            | $674$M  | -       | $512$   | $250$ | -                   | $37$s               | -                | $4.4$G           |
> | LARP-L           | $343$M  | $1024$  | -       | $1024$| $20$s               | -                   | $1.6$G           | -                |
> | OmniTokenizer    | $650$M  | -       | $1280$  | $5120$| -                   | $139$s              | -                | $4.5$G           |
> | NeRV-Diffusion-L | $467$M  | $128$   | $160$   | $250$ | $6.8$s              | $8.2$s              | $1.8$G           | $2.1$G           |

---

> ### Author Response · Authors · 2025-11-26
>
> **W2: More quantitative metrics.**
>
> The use of FVD for quantifying both reconstruction and generation quality on UCF and K600 datasets is a standard pipeline that have been widely practiced by SOTA work in recent top venues including MAGVITs, OmniTokenizer, HPDM, LARP and AR-Diffusion etc.
> They didn't report additional metrics and we initially followed their protocols for fair comparison.
>
> We extend our reconstruction metrics to PNSR, SSIM and LPIPS between the output and input videos, and extend our generation metrics to cross-frame LPIPS (fLPIPS) and C3D/I3D-based Inception Score (IS) on the output videos.
> As suggested by Review WJ85, we also evaluate the non-text subsets of VBench on generators, including Subject Consistency (SC), Background Consistency (BC), Temporal Flickering (TF), Motion Smoothness (MS), Dynamic Degree (DD), Aesthetic Quality (AQ) and Imaging Quality (IQ).
> For comparing methods, we use their pretrained checkpoints for inference.
>
> It worth noting that the ultimate purpose of NeRV-VAE (and the comparing tokenizers) is to encode RGB videos into smooth latents for high-quality video generation, and we optimize it for best generative quality instead of solely reconstruction faithfulness.
> Our models achieve SOTA generation performance on UCF across all dimensions, and reach comparable SOTA performance on K600 frame prediction.
>
> | Reconstruction (UCF) | PNSR$\uparrow$ | SSIM$\uparrow$ | LPIPS$\downarrow$ | rFVD$\downarrow$ |
> |----------------------|--------------:|---------------:|------------------:|-----------------:|
> | TATS                 | $22.43$       | $0.765$        | $0.108$           | $162$            |
> | Open-MAGVIT2 [2]     | $25.84$       | $0.862$        | $0.045$           | $16$             |
> | LARP-L               | $27.87$       | $0.891$        | $0.038$           | $20$             |
> | NeRV-VAE-L           | $26.63$       | $0.879$        | $0.043$           | $22$             |
>
> | Generation (UCF)   | fLPIPS$\updownarrow$ | IS$\uparrow$ | VBench$\uparrow$ | VBench$\uparrow$ | VBench$\uparrow$ | VBench$\uparrow$ | VBench$\uparrow$ | VBench$\uparrow$ | VBench$\uparrow$ | gFVD$\downarrow$ |
> |--------------------|---------------------:|------------:|-----------------:|-----------------:|-----------------:|-----------------:|-----------------:|-----------------:|-----------------:|-----------------:|
> |                    |                      |             | SC               | BC               | TF               | MS               | DD               | AQ               | IQ               |                  |
> | Ground Truth       | $0.031$             | $86.24$     | $0.954$          | $0.977$          | $0.981$          | $0.988$          | $0.282$          | $0.410$          | $0.443$          | $0$              |
> | TATS               | $0.048$             | $68.79$     | $0.910$          | $0.958$          | $0.978$          | $0.980$          | $0.314$          | $0.348$          | $0.437$          | $332$            |
> | VIDM [4]           | -                   | $64.17$     | -                | -                | -                | -                | -                | -                | -                | $263$            |
> | VideoDiffusion     | -                   | $80.03$     | -                | -                | -                | -                | -                | -                | -                | $173$            |
> | LARP-L             | $0.025$             | $68.79$     | $0.955$          | $0.977$          | $0.985$          | $0.991$          | $0.218$          | $0.398$          | $0.393$          | $102$            |
> | NeRV-Diffusion-L   | $0.028$             | $82.17$     | $0.958$          | $0.978$          | $0.983$          | $0.989$          | $0.262$          | $0.392$          | $0.433$          | $97$             |
>
> **(K600 results continue in the next comment.)**

---

> > ### Author Response · Authors · 2025-11-26
> >
> > **(Continued K600 results for W2.)**
> >
> > | Reconstruction (K600) | PNSR$\uparrow$ | SSIM$\uparrow$ | LPIPS$\downarrow$ | rFVD$\downarrow$ |
> > |-----------------------|--------------:|---------------:|------------------:|-----------------:|
> > | LARP-L                | $28.22$       | $0.867$        | $0.035$           | $11$             |
> > | NeRV-VAE-L            | $26.45$       | $0.823$        | $0.044$           | $19$             |
> >
> > | Generation (K600)   | fLPIPS$\updownarrow$ | IS$\uparrow$ | VBench$\uparrow$ | VBench$\uparrow$ | VBench$\uparrow$ | VBench$\uparrow$ | VBench$\uparrow$ | VBench$\uparrow$ | VBench$\uparrow$ | gFVD$\downarrow$ |
> > |---------------------|---------------------:|------------:|-----------------:|-----------------:|-----------------:|-----------------:|-----------------:|-----------------:|-----------------:|-----------------:|
> > |                     |                      |             | SC               | BC               | TF               | MS               | DD               | AQ               | IQ               |                  |
> > | Ground Truth        | $0.031$             | $31.20$     | $0.950$          | $0.945$          | $0.978$          | $0.988$          | $0.266$          | $0.406$          | $0.461$          | $0$              |
> > | LARP-L              | $0.029$             | $23.51$     | $0.933$          | $0.973$          | $0.981$          | $0.989$          | $0.292$          | $0.337$          | $0.293$          | $17$             |
> > | NeRV-Diffusion-L    | $0.034$             | $27.09$     | $0.928$          | $0.967$          | $0.977$          | $0.979$          | $0.354$          | $0.392$          | $0.432$          | $22$             |

---

> ### Author Response · Authors · 2025-11-26
>
> **W3: Limited video resolution and length.**
>
> We scale up NeRV-Diffusion to $256^2$ resolution and $128$ frames on UCF.
> Conventional VAEs encode videos into frame-wise feature maps with a fixed spatial downsampling factor, and thus their latent size increases quadratically w.r.t. to RGB resolutions.
> In contrast, NeRV-Diffusion uses INRs as instance-specific decoders, and we append one additional layer/block to the end to perform an extra upsampling to double the output resolution.
> Therefore, our model parameters increase sublinearly, and the neural weight latent size also only needs to increase accordingly.
> Specifically, our NeRV decoder size increases by $17$% as shown in the decoder table W1, and we increase the neural latent token numbe by $25$% for simple alignment of channel-wise parameterization.
> Besides, our GNeRV incorporates skip connections (Section 3.2) so multi-resolution output and joint training are also feasible.
>
> For video length, conventional VAEs increase their frame-wise latent size linearly with a fixed temporal downsampling factor, while our NeRV features smooth native time interpolation and can be trained with large frame intervals as shown in Section 4.5.1.
> In our $128$-frame experiment, we train NeRV-Diffusion at $16$ frames with the downsampling interval of $8$ and sample it with $8\times$ frame interpolation.
> This would be more challenging as no ground truth frames between the anchored frames are seen by our models.
>
> We report the generation performance comparison in below tables.
> In both cases our NeRV-Encoder remains the same configuration.
> NeRV-Diffusion outperforms recent SOTAs with high compactness and easy extensibility.
>
> | Method                 | gFVD$^{256}$$\downarrow$ |
> |------------------------|-------------------------:|
> | VIDM                   | $263$                   |
> | Latte                  | $202$                   |
> | OmniTokenizer          | $191$                   |
> | AR-Diffusion [3]          | $186$                   |
> | HPDM-M [4]                | $143$                   |
> | NeRV-Diffusion-L (Ours)| $140$                   |
>
> | Method                 | gFVD$_{128}$$\downarrow$ |
> |------------------------|-------------------------:|
> | Latte                  | $1157$                  |
> | DIGAN                  | $1103$                  |
> | PVDM [5]                  | $505$                   |
> | VIDM                   | $426$                   |
> | CoordTok [6]              | $369$                   |
> | NeRV-Diffusion-L (Ours)| $366$                   |
>
> **W4: Dataset scale and complexity.**
>
> UCF and K600 are the common standard benchmarks to evaluate novel video generation architectures, widely adopted by SOTA work in recent top venues including MAGVITs, OmniTokenizer, HPDM, LARP and AR-Diffusion etc.
> Our computation resources in academia are limited and training open-world text-to-video is not feasible.
> NeRV-Diffusion follows the generic diffusion pipeline and it can be extended to additional condition input (please refer to **W7**).
>
> Meanwhile, UCF dataset contains over $13$K real-world videos sourced from YouTube of average $180$ frames, totally about $27$ hours.
> K600 dataset contains about $500$K videos of average $250$ frames, summing up to about $57$ days in length.
> The total information is far more than our models' parameters.
> They are realistic and complex data and training and evaluating on them are non-trivial.
>
> **W5: Latest comparison methods and metrics.**
>
> We are comparing our models with latest SOTA methods.
> For example, AR-Diffusion was published on CVPR 2025, and LARP was published on ICLR 2025.
> NeRV-Diffusion outperforms them without longer training.
> Wan2.1 is a large-scale commercial text-to-video foundation model, which is out of our resource scope and training from scratch is not feasible in academia.
> Self-Forcing is an add-on approach to finetune pretrained video diffusion models for long video roll-out, while our work focuses on a new architecture of video diffusion models and they are two different tracks.
>
> We extend our metrics for both reconstruction and generation and our models outperforms previous SOTAs (please refer to **W2**).
> We also compare the inference efficiency among our and previous SOTA methods and NeRV-Diffusion leads by a large margin (please refer to **W1**).
>
> **W6: Qualitative comparisons.**
>
> We add additional qualitative comparison with LARP in our revised paper in Appendix Figures A4 and A5.
> We sample both models with the same class label input.
> We also embed the raw videos into the revised PDF in Appendix Figure A6.
> Please use Adobe Acrobat Reader to open on laptop/desktop and clck to play.
> Compared to LARP, NeRV-Diffusion also constructs holistic video representations but meanwhile still maintain the spatiotemporal integrity via the query time indices and spatial input embeddings, and thus produces more structural videos with less morphing or tearing.

---

> ### Author Response · Authors · 2025-11-26
>
> **W7: Quantitative verification of interpolation, and generalization to more tasks.**
>
> Reviewer WpVC also asked quantitative verification of long video interpolations, and we follow their suggestions to quantitatively measure semantic consistency in terms of object identity and action logic preservation.
> Specifically, we leverage the Subject Consistency and Background Consistency metrics in VBench, which extract the subject and background features via DINO/CLIP, and calculate their pairwise similarities across all frames for appearance preservation.
> Inspired by it, we further employ C3D, a action recognizer pretrained on short UCF videos, and feed it with all sub-clips from the long videos to extract action features.
> We calculate pairwise similarities of the action features across all windows to measure the action logic preservation.
> These metrics don't only rely on consecutive distance but average along the whole video, accurately reflecting the drifting issues for long video generation.
>
> | Long Consistency | VBench$\uparrow$ | VBench$\uparrow$ | Action Sim.$\uparrow$ |
> |------------------|-----------------:|-----------------:|----------------------:|
> |                  | SC               | BC               |                        |
> | Ground Truth     | $0.931$          | $0.956$          | $0.912$               |
> | Ours             | $0.919$          | $0.942$          | $0.901$               |
>
> Our implicit diffusion model follows standard diffusion framework but only switches to neural weight latents, and thus can be trained with various condition types.
> Additional condition input can be integrated with standard cross attention so that different modalities other than similar videos or images are supported.
> We follow the prior work that developed novel tokenizers to majorly test on the foundation tasks.
> Due to time limit and resource scale, we extend NeRV-Diffusion to several additional tasks, including image-to-video generation and unconditional video generation.
> As suggested by Reviewer FaBD, we also include an edge-to-video generation experiment.
> Previous work didn't evaluate these protocols, and we present our qualitative results in the revised paper Appendix Figure A7.
>
> **W8: Novelty and contribution compared to DIGAN.**
>
> GANs and diffusions are two distinct generative model families.
> The idea of generating INR weights has emerged before DIGAN and has been developed for various modalities with different model designs, and we have discussed them in related work.
> Our proposed modules such as multi-head affine and channel-wise parameterization has effectively boosted the performance of our models to achieve the first implicit video diffusion model, and we have depicted the impact of key design options in ablation studies.
> In strength you also mentioned that "applying diffusion models to video INR weight space still has exploratory significance".
> Reviewer TaBD also describes our work as "a new paradigm", and Reviewer WpVC marks our work as "highly innovative".
>
> **References**:
>
> [1] Efficient Video Diffusion Models via Content-Frame Motion-Latent Decomposition. ICLR 2024.
>
> [2] Open-MAGVIT2: An Open-Source Project Toward Democratizing Auto-regressive Visual Generation. ArXiv 24.09.
>
> [3] AR-Diffusion: Asynchronous Video Generation with Auto-Regressive Diffusion. CVPR 2025.
>
> [4] Hierarchical Patch Diffusion Models for High-Resolution Video Generation. CVPR 2024.
>
> [5] Video Probabilistic Diffusion Models in Projected Latent Space. CVPR 2023.
>
> [6] Efficient Long Video Tokenization via Coordinate-based Patch Reconstruction. ArXiv 24.11.

---

> > ### Author Response · Authors · 2025-11-26
> >
> > If you find our response helpful, please kindly consider raising your ratings.
> > If you have more concerns, please feel free to let us know and we will promptly update.
> > We sincerely appreciate your valuable discussion and feedback!

---

### Author Response · Authors · 2025-12-03
**Summary of Reviews and Responses for the Area Chair**

In a year of exceptional challenges, we greatly appreciate the Area Chairs’ expanded involvement in the discussion period. Thank you for the considerable time and effort you have invested in our submission.

Below we summarize the highlights and concerns of reviewers, as well as our key responses and revisions to the manuscript. We have extensively updated the PDF to address all concerns, mainly in the Appendix Section E, maked with light blue background.

## Highlights from reviewers

The idea of shifting from generating pixels to generating neural network weights is **“unique in the video generation field”** (Reviewer kMwA), establishing **“a new paradigm”** (Reviewer FaBD) and is **“highly innovative”** (Reviewer WpVC).
The framework is **“compact and lightweight"** in all the decoder, latent and generator components with **"high inference efficiency"** (Reviewers FaBD, WJ85).
Key architectural improvements are **"solid"** (Reviewer WpVC) and have demonstrated **"high-quality results"** (Reviewer FaBD), **"well-supported"** by ablations (Reviewers WJ85, WpVC).
The holistic video INR generation enables **"smooth frame interpolation"** with good temporal coherency (Reviewers FaBD, WJ85).

## Key concerns and questions from reviewers

4 reviewers raise 6 common concerns and questions, and we have resolved them with new experiments and evaluations and significantly elevated our contribution.

- **Inference efficiency comparison (Appendix Section E.1):** We have compared the inference efficiency with other SOTA works as well as commercial foundation models. Our models cost far less latency and VRAM footprint in both decoding and generation stages.

- **Scaled up video resolution and length (Appendix Section E.2):** We have conducted new experiments on higher resolution ($256^2$) and longer videos ($128$ frames). Our models feature sublinear increase in parameters and efficiency w.r.t. resolution and length increase. Our model outperforms recent SOTAs with high compactness and easy extensibility.

- **Additional quantitative metrics  (Appendix Section E.3):** We have extended our evaluation metrics for both reconstruction and generation tasks. Our models achieve SOTA generation performance across all benchmarks while maintaining a low recontruction-generation gap, indicating the effectiveness of our implicit latent framework.

- **Quantitative verification of interpolation  (Appendix Section E.6):** We have adopted new measurements over the whole video duration to verify the global consistency of our long video interpolation in both appearance and action preservation.

- **Control of granular conditions in the implicit space  (Appendix Section E.7 & E.11):** Our model follows the standard diffusion framework and supports all kinds of conditions. We have extended it to additional setups including unconditional video generation, image-to-video generation and edge-to-video generation, exhibiting its versatility.

- **Two-stage models and balance between reconstruction and generation  (Appendix Section E.9 & E.10):** We have explained that all modern large-scale generation models adopt a two-stage framework (e.g. KL-VAE, VQGAN), where raw data are first encoded into a latent space and the generator is trained in this latent space. The trade-off between the encoding compactness and reconstruction expressiveness is common in all two-stage frameworks. We are able to manage this balance well to reach high generative performance.

## Other concerns and questions from reviewers

We have also addressed other concerns and questions from reviewers via new experiments, evaluations and discussions, both in the individual comments and in the revised **Appendix Sections E.1 - E.22, Tables A2 - A15, and Figures A4 - A8**.
We also appreciate Reviewer WJ85's intention to raise our score.

---

### Meta-Review · Area_Chair_ihxp · 2026-01-07

**Summary:**

This paper proposes a unique approach for video synthesis via diffusing implicit representation. This approach presents advantages in generation efficiency and seamless resizing. Experimental results, including the ones from the rebuttal, demonstrate its effectiveness in both speed and quality. The additional results should be incorporated into the final version to provide a more holistic view of the method for further research.

**Reviewer Concerns:**

Summary in https://openreview.net/forum?id=tX0cSOvBnS&noteId=RNqzYxA8Kr is mostly accurate.

**Reviewer Scores:**

WpVC: 6 -> 6
FaBD: 6 -> 6
WJ85: 4 -> 4
kMwA: 2 -> 4
Average: 4.5 -> 5

---

### Decision · Program_Chairs · 2026-01-26

Accept (Poster)